# Cu- or Ag-containing Bi-Sb-Te for in-line roll-to-roll patterned thin-film thermoelectrics

Xudong Tao [1] ✉, Qianfang Zheng[1], Chongyang Zeng[2], Harry Potter[1], Zheng Zhang[1], Joshua Ellingford [3], Ruy S. Bonilla [1], Emiliano Bilotti[2], Patrick S. Grant[1] & Hazel E. Assender [1]

The Selective Metallization Technique shows promise for roll-to-roll in-line patterning of flexible electronics using evaporated metals, but challenges arise when applied to sputtering functional materials. This study overcomes these challenges with simultaneous sputtering of Bi-Sb-Te and evaporation of metal (Ag or Cu) for thermoelectric layers when using Selective Metallization Technique. Large-scale manufacturing is demonstrated through roll-to-roll processing of a 0.8 m wide polymer web at 25 m/min, achieving high-throughput production of functional thin-film patterns with nanometer thickness. The room-temperature-deposited material system exhibits significantly enhanced thermoelectric performance and facilitates an n-type-to-p-type transition in the Cu- or Ag-containing Bi-Sb-Te-based composite film. Here, we show that while applying Selective Metallization Technique, the evaporation of metal modifies the impact of residual oil on Bi-Sb-Te, which can be effectively removed with a few seconds of plasma exposure, and the fabricated thermoelectric devices are validated in wearable applications utilizing a coiled-up wristband design.

Exploring in-line patterning tools for large-scale manufacturing of flexible thin-film electronics is particularly timely in the era of the Internet of Things[1]. Recent advances in the Selective Metallization Technique (SMT)[2–4] show promise over conventional patterning techniques, such as printing, which is generally restricted to micrometer-range thicknesses[5,6]. However, certain materials, like thermoelectric materials, benefit from nanostructure to optimise performance[7,8], and nanometer-thick thin films reduce concerns related to material cracking[9], which is crucial for flexible electronics applications. SMT is a nanometer-thick in-line patterning technique using roll-to-roll (R2R) processing. It employs a flexographic technique to print oil patterns on a flexible polymer web, which then passes over a vapor source (e.g., evaporation) to remove oil patterns and simultaneously deposit nano-thick material patterns. We have successfully demonstrated the integration of SMT with evaporation for the R2R manufacture of metal patterns with nanometer thickness and micrometer resolution[2–4]. However, challenges persist when applying SMT with sputtering for functional or multi-element materials. For example,

with Bi-Te-based thermoelectric thin films[4], the oil used in SMT degraded the functional material due to oil-induced doping/oxidation and significant oil residuals. Building upon the success of SMT with evaporation, herein we propose a straightforward solution: introducing a metal evaporation source during the sputtering of functional materials in SMT to modify the effect of the oil on the materials.

Functional thermoelectrics in thin-film form show promising potential as local power sources for low-power wearable and implantable electronics[10,11]. Bismuth telluride-based alloys[12,13] are the most common thermoelectric materials operating at temperatures near that of the human body[12]. Bismuth antimony telluride (Bi-Sb-Te) is regarded as an ideal p-type thermoelectric material with an excellent thermoelectric figure of merit[14–16]. Previous studies[17–19] have shown that doping bulk Bi-Sb-Te can significantly improve thermoelectric performance. For example, Cu atoms could reside between the loosely bonded Te layers in the crystal lattice, acting as a rapid diffuser and contributing to electrical and electronic thermal conductivity[20]. Similarly, Ag atoms can occupy Bi/Sb sites, increasing p-type carrier (hole) concentration and

[1]Department of Materials, University of Oxford, Parks Road, Oxford, UK. [2]Department of Aeronautics, Imperial College London, Exhibition Road, London, UK. [3]Plasma Quest Limited, Unit 1B, Rose Estate, Osborn Way, Hook, Hampshire, UK. ✉e-mail: xt240@cam.ac.uk

effectively suppressing bipolar excitation, leading to a higher power factor (*PF*) and figure of merit[19]. These findings from bulk studies inspire research into the thin-film configurations of doped Bi-Sb-Te, which exhibit outstanding opportunities for flexible electronics applications.

Herein, we combined the sputtering of Bi-Sb-Te with the evaporation of Cu or Ag to explore the effect of metal sources in a semiconductor thin-film system. This process was conducted in an R2R coating facility that mirrors typical industrial manufacturing capability using our SMT technique, aiming to minimize the deleterious effect of the oil in SMT when compared to sputtering alone. We successfully demonstrated that the SMT-fabricated Cu-Bi-Sb-Te exhibited excellent thermoelectric performance. The performance of the fabricated thin-film thermoelectrics was validated using a coiled-up design[21], and we further integrated the coiled-up cells into a wristband for wearable applications. This R2R-type SMT technique makes significant progress in patterning high-performance nano-thick functional materials in an industrial setting for large-scale manufacturing, using a 0.8 meter wide polymer web at a high throughput of 25 m min⁻¹ in-line speed.

## Results and discussion
### SMT in-line patterning of functional materials
The SMT process (Fig. 1) was operated in a mode to mimic a continuous R2R process. A 0.8 m wide flexible polymer substrate revolved around a rotating coating drum at an in-line speed of 25 m min⁻¹. The flexography part contacted the polymer web to print oil patterns. After the flexography component was retracted, the polymer web with oil patterns passed through the deposition sources multiple times for layer-by-layer sputtering of Bi-Sb-Te and simultaneous evaporation of Cu or Ag. The metal feed-rate was varied from 1 to 4, with the number 20 corresponding to an in-line speed of 8 cm min⁻¹ (see Methods for details). The process was analogous to lithography, where the oil patterns act as photoresist patterns, followed by thin-film deposition for metal patterning. Similar to the lift-off process used to remove photoresist residuals in lithography, the oil used in SMT was cleaned by R2R-compatible processing such as e-beam or plasma. Hence, a lithography-like process was achieved in R2R processing. This technique enables high-throughput in-line patterning of nanometer-thick metal and functional materials in an industrial R2R setting, achieving micron resolution and high-density arrays, as demonstrated in Fig. 1 and Supplementary Movie 1.

### 2.2 Cu- or Ag-containing Bi-Sb-Te materials
**Film thickness, surface morphology and elemental composition.** A control group of continuous thin films was fabricated by simultaneous sputtering of Bi-Sb-Te and evaporation of Cu (or Ag) onto flexible polymer substrates mounted on a rotating coating drum to optimize the metal content. The films' granular morphology was evident in atomic force microscopy (AFM) images (Fig. 2a) and scanning electron microscopy (SEM) images (see Figure S1a, Supplementary Information). With an increased fraction of evaporated metals (Ag or Cu), there was a noticeable rise in root mean square roughness and grain size (Fig. 2a, b), resulting in fewer grain boundaries and, consequently, a reduced scattering effect on electrical performance[22]. As expected, the film thickness broadly increased with greater exposure to the metal source, ranging from 80 nm to 180 nm (see Figure S1b, Supplementary Information). The increase in thickness is attributed to the higher metal feed-rate, as the deposition time was fixed at 5 minutes for all samples. In conventional co-sputtering or co-evaporation systems, vapor sources are typically positioned on the same side, directing material towards the substrate simultaneously. In contrast, this sputtering-co-evaporation system employs spatially separated vapor sources, with the sputtering and evaporation sources located on different sides (see Fig. 1). The substrate, mounted on a rotating coating drum, alternately passes through the sputtering and evaporation zones, resulting in a 'layering' effect with alternating thin layers of sputtered and evaporated materials deposited during each pass. Given the drum circumference of 1.8 m and a rotation speed of 25 m/min, the substrate passed through the deposition sources approximately 69 times over 5 minutes. From this, and the known deposition rate of a sputtered-only sample, the average metal evaporation thickness per pass can be estimated to be sub-nanometer-thick layers per pass – at most 1.5 nm for the greatest feed rate (see Figure S1d, e, Supplementary Information). At such small scales, particularly at low metal feed-rate, the metal is unlikely to form a continuous and uniform layer-by-layer structure, especially given that the deposition occurs at room temperature. Instead, some degree of interdiffusion (e.g., doping), local accumulations (e.g., nucleation, dewetting, phase formation), and a heterogeneous structure at the nanoscale are expected (see Figure S1f, Supplementary Information). In the form of the sub-nanometer-thick alternating layers, the metal and Bi-Sb-Te regions are indistinguishable due to their uniform distribution, as confirmed by energy-dispersive X-ray spectroscopy (EDX) mapping (see Figure S1a, Supplementary Information). At room-temperature deposition, the low mobility of sputtered adatoms and the high mobility of evaporated adatoms further complicate the material's structure. Broadly, regions where sputtered Bi-Sb-Te is combined with metal evaporation exhibit high conductivity, whereas sputtered-only regions, primarily Bi-Sb-Te, demonstrate poor conductivity. Characterizing these regions via SEM (cross-sectional view) and X-ray diffraction (XRD, phase identification) is challenging due to the thinness of the film. This heterostructure structure, with its varying-conductivity geometry, could induce an energy filtering effect, a crucial mechanism for enhancing the performance of thermoelectric thin films. As the metal feed-rate increases further (a maximum of 4 in this study), metal islands may form a continuous layer, causing the materials to behave more like a metal.

The film was too thin for sufficient signal detection from XRD for phase identification (see Figure S2, Supplementary Information). The elemental composition of the Bi-Sb-Te-based composite films was determined through EDX, shown in Fig. 2c. The metal fraction increased with the wire feed-rate for evaporation, as anticipated. While the atomic% of Bi remained broadly consistent, the elemental ratios of Te and Sb decreased with increasing metal evaporation. This phenomenon was attributed to a re-evaporation process caused by the hot evaporation source and/or radiation heat from the evaporation boat. The preferential removal of Sb and Te elements (see Figure S1c, Supplementary Information) relates to their small atomic mass/size and vapor pressure. The etching phenomenon also explained the slight decrease in thickness (compared to the pristine Bi-Sb-Te sample) of the 1Ag and 1Cu films. The alteration in elemental composition was expected to significantly impact the thermoelectric performance of Bi-Sb-Te[23] by influencing carrier concentration and the type of charge carriers[24–26].

**Thermoelectric characteristics.** In terms of the Seebeck behavior, the application of the metal source changed the type of Bi-Sb-Te semiconductor from n-type to p-type, as indicated by the sign of Seebeck coefficient (*S*) in Fig. 3a, where a positive value indicated p-type and a negative value indicated n-type. Although $Bi_{0.5}Sb_{1.5}Te_3$ was expected to be p-type, the sputtered Bi-Sb-Te film here was n-type, signifying that electrons were the majority charge carriers. The n-type Bi-Sb-Te has also been reported previously[27–30]. This behavior could be attributed to the elemental composition of the as-sputtered Bi-Sb-Te (0.5: 1.5: 3.2, in this study), particularly the excess Te leading to Te anti-site occupancy in Bi sites[27]. The electron-dominant n-type Bi-Sb-Te transformed into the hole-dominant p-type Ag- or Cu-containing Bi-Sb-Te by adding the evaporated metal source. Consequently, the system encompassed two types of carriers that determined the sign of *S*.

$$S = \frac{S_e \sigma_e + S_h \sigma_h}{\sigma_e + \sigma_h} \qquad (1)$$

where $\sigma_h$ and $S_h$ are the electrical conductivity and Seebeck coefficient for holes, the $\sigma_e$ and $S_e$ are the electrical conductivity and Seebeck coefficient for electrons ($S_e < 0$, $S_p > 0$).

In contrast to previous studies[31–37] where the Ag(or Cu)-doped Bi-Sb-Te remained p-type, this study expanded the doping range from n-type to p-type because the initially fabricated Bi-Sb-Te was n-type.

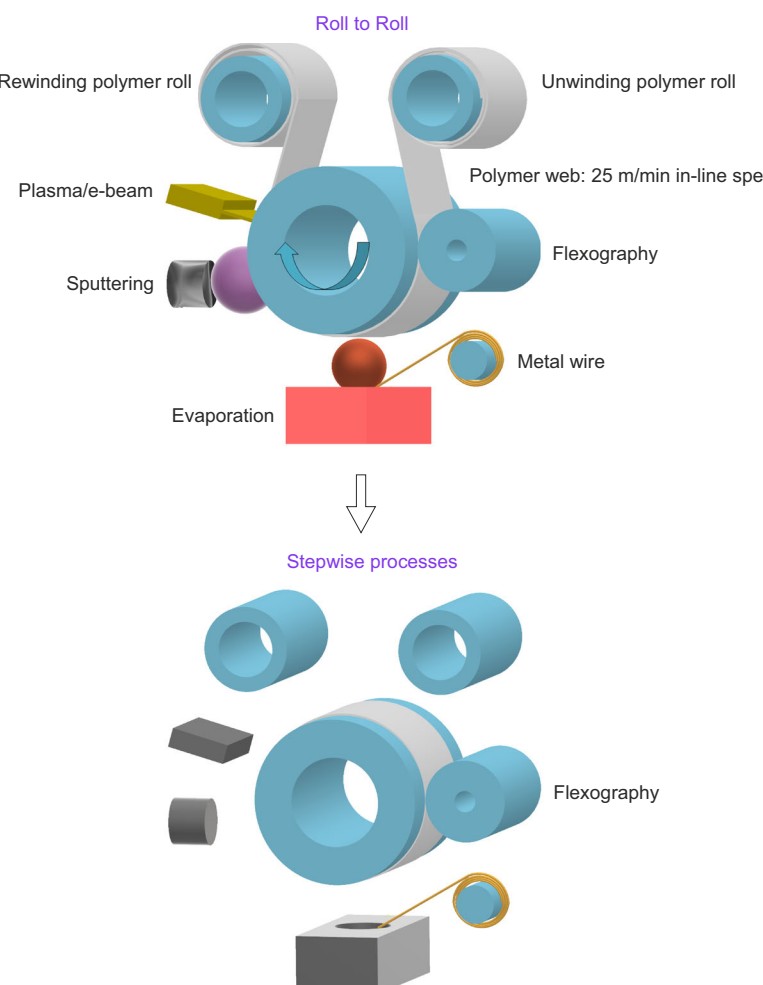

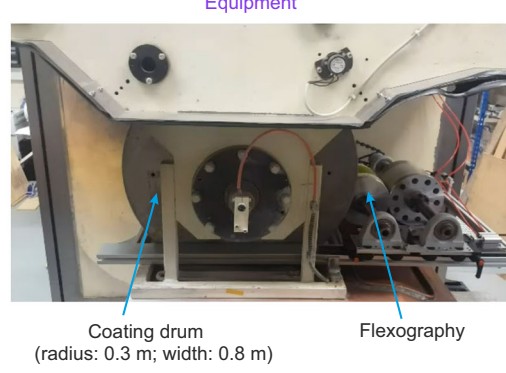

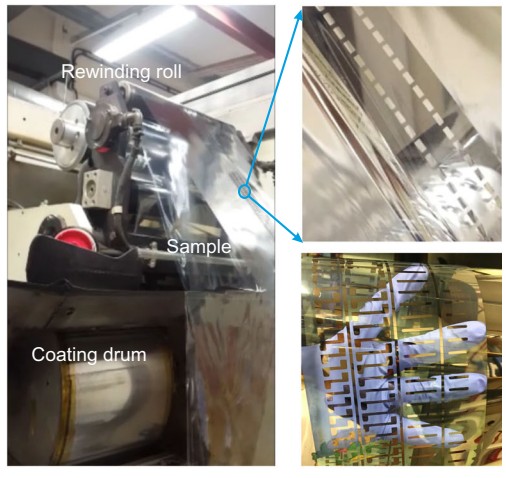

Step 1: The flexography process applies oil patterns on the polymer web

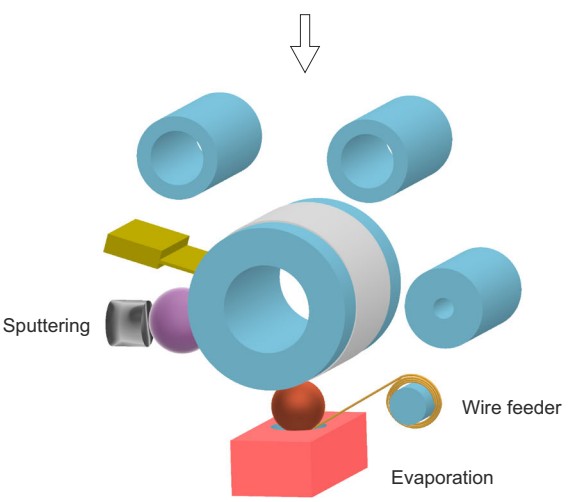

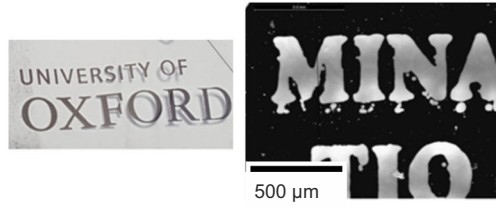

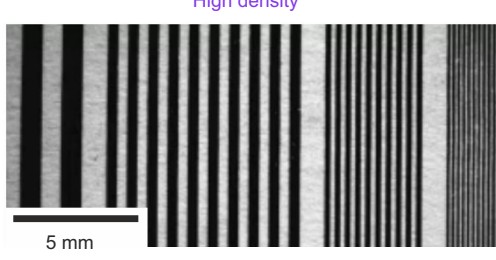

Step 2: The oil-patterned polymer web passes through vapor sources for material deposition

**Fig. 1 | Lithography-like R2R processing for high-throughput manufacturing of flexible thin-film micron-patterns.** In-line patterning of flexible thin films at high throughput by simultaneous sputtering of functional materials and evaporation of metal in an SMT. In evaporation, a feed-rate of 20 corresponded to an in-line speed of 8 cm min⁻¹ for ~1 mm diameter metal wires. The coating drum was rotated at an in-line speed of 25 m min⁻¹. The images show the setup of flexography and the coating drum, demonstrating the large-scale capability for achieving micro-resolution and high-density arrays on flexible thin polymer substrates.

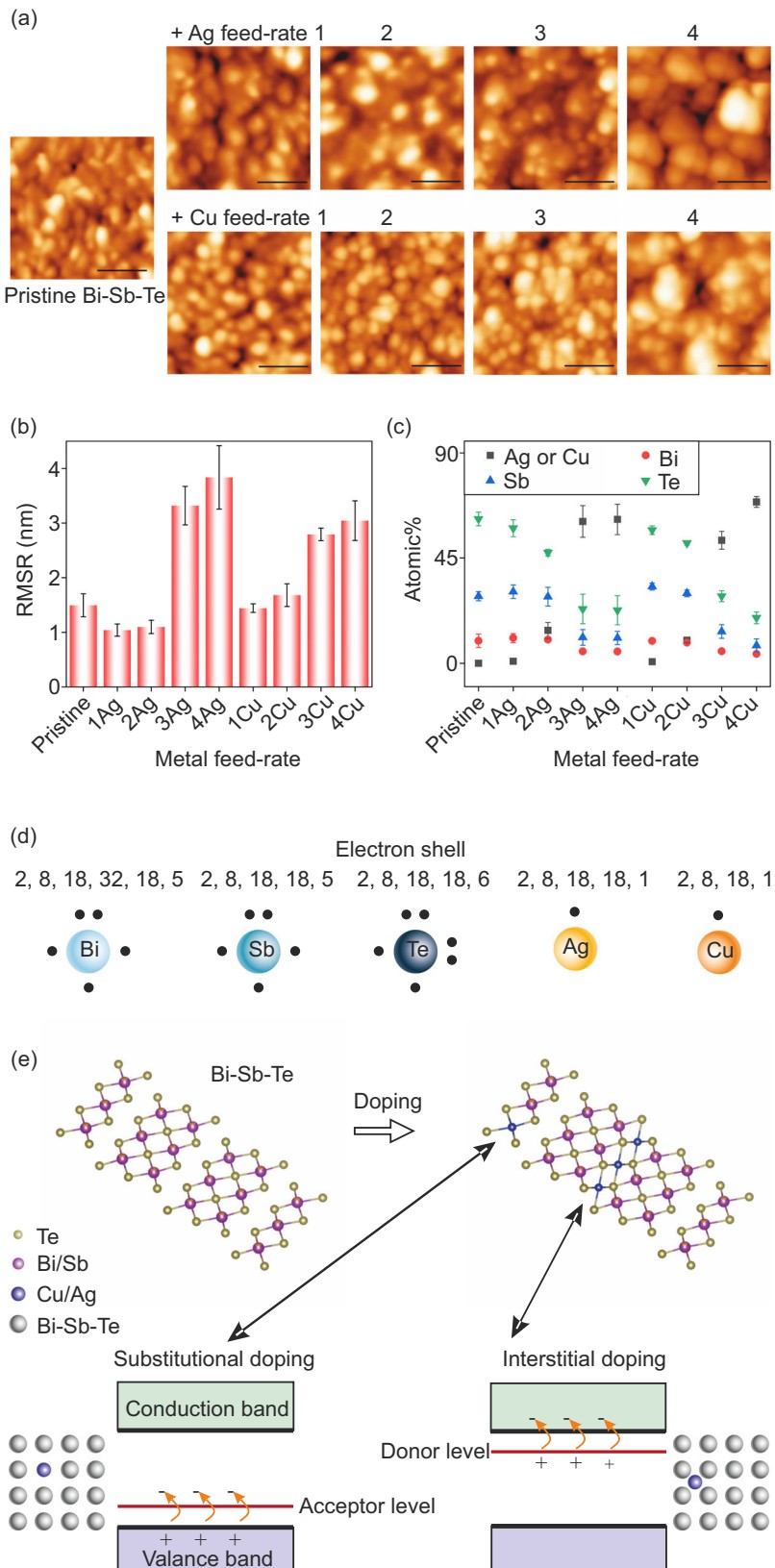

The doping of Ag or Cu could involve both substitutional and interstitial doping[38], corresponding to hole donor or electron donor (contributing to $S_p$ or $S_e$, respectively, see Eq. 1). Both dopants (Ag and Cu) have one valence electron, which is much less than that of Bi (5 electrons), Sb (5 electrons), and Te (6 electrons), see Fig. 2d. If substitutional doping occurs, at least four holes would be provided to the semiconductor system, i.e., a p-type dopant. If interstitial doping occurs, the valence electron of Ag or Cu would be provided to the semiconductor, i.e. an n-type dopant. Regarding the doping of Ag and Cu in the Bi-Sb-Te matrix, both interstitial and substitutional doping could occur, involving interstitial defects located between quintuple layers and substitution at Bi/Sb sites[19,31,33,39], see Fig. 2e.

**Fig. 2 | Ag- or Cu-containing Bi-Sb-Te films: surface morphology, elemental composition, and doping mechanism. a** AFM images of evaporated-sputtered films with varying concentrations of metal (scale bar = 100 nm). **b** Surface root mean square roughness - RMSR (Data are presented as mean ± standard deviation from measurements at nine locations on three identical samples). **c** Elemental composition from EDX-SEM of Bi-Sb-Te composite films (Data are presented as mean ± standard deviation from measurements at eight locations on three identical

samples). **d** Electron shells of these elements with valance electrons. **e** Interstitial & substitutional doping/alloying, including a visualization of the crystal structure and a band diagram showing anticipated acceptor/donor levels. Blue spheres represent the substitutional or interstitial atoms within the host lattice (grey spheres). Acceptor levels introduce more '+' (holes) into the valence band, while donor levels contribute more '-' (electrons) to the conduction band. The crystal structure models are generated in VESTA using the pure Bi-Sb-Te source data (ICSD: 147547).

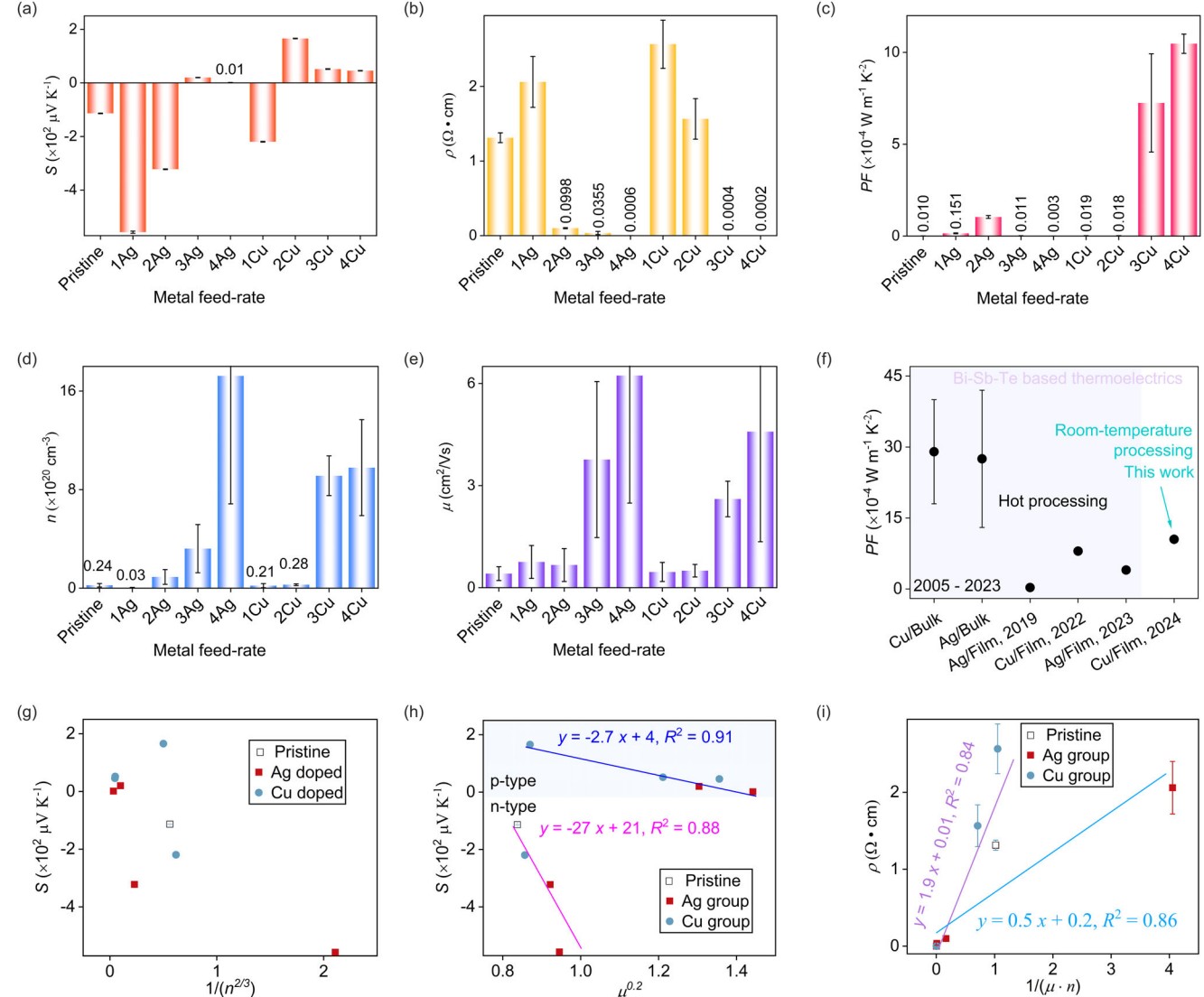

**Fig. 3 | Thermoelectric characterization of Ag- or Cu-Containing Bi-Sb-Te Films. a–e** Effect of metal feed-rate on (**a**) $S$ – Seebeck coefficient; (**b**) $\rho$ - Electrical resistivity (Data of $S$ and $\rho$ are presented as mean ± standard deviation based on nine measurements); (**c**) $PF$ - Power factor; (**d**) $n$ – Carrier concentration; (**e**) $\mu$ - Carrier mobility (Data of $n$ and $\mu$ are presented as mean ± standard deviation based on measurements from three identical samples). **f** Comparison of Cu- or Ag-containing Bi-Sb-Te materials: 'Cu/Bulk' refers to a Cu-containing Bi-Sb-Te bulk sample, while

'Ag/Film' or 'Cu/Film' refers to Ag- or Cu-containing Bi-Sb-Te thin-film samples (see references in Table S3, Supplementary Information). **g** Seebeck coefficient vs (carrier concentration)$^{-2/3}$. **h** Seebeck coefficient vs (carrier mobility)$^{0.2}$ (the lines indicate the linear fits for both the n-type and p-type regions). **i** Electrical resistivity vs (carrier concentration*mobility)$^{-1}$ (the lines indicate the linear fits for both the Ag and Cu groups).

Figure 3a shows the distinctive n-type to p-type transition. The reasons for this transition are summarized in Table 1, noting that all the listed phenomena might happen simultaneously, but some might dominate. The pristine Bi-Sb-Te was n-type due to the excess of Te, acting as an electron donor. Based on the limited data points in our case, the behavior of 2Ag and 2Cu films was at the boundary for the n-type-to-p-type transition. Further resolution of this boundary regarding the metal content was not explored due to limitations in the wire feed-rate of our fabrication system. The following sections provide a

detailed discussion regarding this type of transition phenomenon. ('n' in nAg and nCu denotes the feed-rate of the metal source during evaporation. A higher feed-rate indicates a greater metal content in the Bi-Sb-Te system).

**Metal feed-rate of 0 - 1.** At this stage, the material remains n-type. With few Ag and Cu dopants, the fraction of Bi and Sb remains almost constant (see Fig. 2c), but the Te fraction drops sharply, resulting in a decrease in carrier (electron) concentration. As a result, $S$ increases in

**Table 1 | Analysis of n-type-to-p-type transition ($S_e$ values are negative, and $S_p$ values are positive. '↑' indicates an increasing trend, while '↓' indicates a decreasing trend)**

| S behavior | | Majority carrier | Dominant phenomena | Results | |
|---|---|---|---|---|---|
| n-type | \|S\| ↑ | Electron | The decrease in Te content; The heterostructure interface | Electron concentration ↓ | Negative S. $\|S_e\|$ ↑ so \|S\| ↑ |
| | \|S\| ↓ | | The interstitial doping | Electron concentration ↑ | Negative S. $\|S_e\|$ ↓ so \|S\| ↓ |
| p-type | Transition appears | Hole | The substitutional doping; Formation of other phases | Hole concentration dominates | Positive S. $\|S_p\| > \|S_e\|$ |
| | \|S\| ↓ | | | Hole concentration ↑ | Positive S. $\|S_p\|$ ↓ so \|S\| ↓ |

the n-type region due to the inverse relationship between $n$ and $S$[40,41]. As previously discussed, a varying-conductivity geometry (see Figure S1f, Supplementary Information) could form heterostructure interfaces and induce an energy filtering effect[33] - a common method used to modify carrier properties and thereby enhance $PF$ in thin-film thermoelectrics[31]. Carriers preferentially traverse the highly conductive regions, while the poorly doped regions with low conductivity act as energy filters at the boundaries. Carriers with low energy are blocked at the boundaries, whereas high-energy carriers pass through, resulting in reduced $n$ but increased $\mu$, thereby increasing $S$ (see Fig. 3a, d, e for the 1-metal case).

**Metal feed-rate of 1 - 2.** The material remains n-type. With an increased amount of metal source, interstitial doping could dominate, introducing donor levels close to the conduction band (see Fig. 2e), thus increasing $n$ (Fig. 3d). Given the inverse relationship between carrier concentration and Seebeck performance[40,41], the $S$ decreases at this stage.

**Metal feed-rate at 2 or 3.** The semiconductor starts to transition from n-type to p-type, indicating that the majority of carriers are holes. This shift could be due to a change in the dominant doping mechanism from interstitial to substitutional, or the formation of other material phases. The coexistence of interstitial and substitutional doping has been observed in other material systems[42–44] and the doping type can be influenced by heat treatment[43]. Our fabrication approach includes heat treatment from the evaporation source. Although changes in doping type in the doped Bi-Sb-Te system have not been previously reported in the literature, it is known that Cu or Ag can be either interstitial or substitutional dopants in the Bi-Sb-Te system[39], see Fig. 2e. It can be assumed that at low dopant concentrations, dopant atoms can easily occupy interstitial sites without significant lattice distortion. However, at higher dopant concentrations, interstitial sites become saturated, thus the crystal lattice cannot accommodate more atoms in the interstitial positions without significant lattice distortion. Consequently, the dopant atoms may start replacing host atoms, such as Bi or Sb, acting as substitutional dopants. In addition, the decreasing amounts of Sb and Te, as shown in Fig. 2c also affect the doping mechanism, if considering these elements as the original dopant sources in the pristine Te-excess Bi-Sb-Te system. It should be noted that other phases, such as the metal-dominated thermoelectric material $Cu_2Te$, could form[45–47] when a significant amount of metal source and heat treatment by the evaporation boat are involved. They might form an integrated system, such as (Cu-Te)(Bi-Sb-Te)[45], or the Cu-Te phase could become a significant separate phase.

**Metal feed-rate > 3.** The material remains p-type as holes are the majority carriers. As the metal source increases, substitutional doping will introduce acceptor levels near the valence band (Fig. 2e). These levels accept electrons from the valence band, leaving more holes in the valence band that contribute to electrical conduction (Fig. 3b, d). As the hole concentration increases, the $S$ decreases. At much higher metal content, a pure metal phase could form as inclusions within the

thermoelectric matrix[48], which could significantly contribute to thermoelectric performance by enhancing $S$ through energy filtering and/or reducing $\rho$ via carrier channeling or injection. Alternatively, metal-rich thermoelectric phases e.g. $Cu_2Te$[46,49], $Cu_4Sb$[50], and $Ag_2Te$[51–53] could contribute to the performance of the alloy system. These phases may form due to the annealing effect induced by the substantial amount of hot metal source and the hot evaporation boat. The material would consist of multiple phases, such as crystalline/amorphous regions of $Cu/Bi$-$Sb$-$Te/Cu_2Te/Cu_4Sb$. This would introduce a significant number of boundaries, altering scattering mechanisms and affecting carrier transport properties. The energy filtering effect may occur, where the highly conductive phases decrease $\rho$ while less conductive regions serve as energy filters, increasing $S$. Hence, the formation of metal-rich phases and pure metal inclusions can have a complex yet potentially beneficial effect on thermoelectric performance, resulting in a notably high $PF$ for the 4Cu sample (see Fig. 3c).

Based on the analysis above, the transition from interstitial to substitutional doping could explain the n-type-to-p-type transition observed at a metal feed-rate at 2 or 3. This transition likely leads to a shift in the Fermi level, which initially rises and then falls as the dopant concentration increases[54]. This analysis is based on a simplified model of the doping mechanism within the Bi-Sb-Te matrix (Fig. 2e). However, the actual situation is more complex, particularly due to our room-temperature fabrication process (resulting in poor crystallinity, as confirmed in our previous works[55,56]), annealing effects from the hot evaporation boat/vapor source, and the heterogeneous structure during layer-by-layer deposition. Additionally, metal-rich thermoelectric phases could form due to the significant amount of metal source involved. These factors result in variations in elemental composition, grain size and crystallinity as well as the coexistence of multiple phases, creating a varying-conductivity geometry. We can confirm the presence and uniform distribution of metal, Bi, Sb, and Te elements through SEM/EDX (see Figure S1a, Supplementary Information), however, identification of the multiple possible phases is not feasible in our case because the film thickness is insufficient for XRD detection (see Figure S2, Supplementary Information). This limitation is further compounded by the room-temperature R2R deposition technique, which restricts film crystallinity. We selected this thickness range because rapid R2R thin-film deposition becomes less favorable for thicker films; the very thin films have the manufacturing advantage of high deposition throughputs, to keep down the manufacturing cost. The nanometer thickness of some functional materials, for example thermoelectric materials[7,8], is particularly advantageous for enhancing performance. Our prior work has demonstrated that films of this thickness can achieve good thermoelectric performance[56]. This thickness range is compatible with the SMT technique, which is constrained by the thickness of the oil patterns, although the oil thickness can be adjusted in flexography.

In Fig. 3d, the change in $n$ aligns with the regimes described above. However, the expected linear relation[57] between $S$ and $1/(n^{2/3})$ was not as clear as anticipated in Fig. 3 (g). As expected, $\mu$ keeps increasing with higher metal content (Fig. 3e). $\mu$ also influences $S$, where $S$ is proportional to $S \propto \mu^{0.258}$. There were robust linear relationships with an $R^2$

value of ~0.9 between $S$ and $\mu^{0.2}$ for both the n-type and p-type regions (Fig. 3h). This correlation could be attributed to the growth of grain size (i.e. less scattering effect) and/or the change in elemental composition (i.e. doping effect), as shown in Fig. 2. $\rho$ can be interpreted in terms of $n$ and $\mu$; linear relationships, with an $R^2$ value of ~0.85, were observed for both the Ag group and the Cu group in Fig. 3i. Similarly, $\rho$ was affected by the phenomena outlined in Table 1. At a low concentration of metal, the heterostructure interface might introduce a carrier energy filtering effect that only allowed the high-energy carrier to pass through (i.e. high $\mu$ and less $n$). Furthermore, fewer carriers (lower $n$) would decrease collisions between carriers, thereby increasing $\mu$. At a high concentration of metal, substitutional doping dominates, with Ag or Cu replacing the Bi or Sb atoms in their crystal sites[31], resulting in a significant increase in both $\mu$ and $n$, and consequently in $\rho$.

$PF$ depended on both $S$ and $\rho$ with an increase in $PF$ observed in the n-type region for the Ag-containing Bi-Sb-Te (e.g., the 2Ag sample, $1.0 \pm 0.1 \times 10^{-4}$ W m$^{-1}$ K$^{-2}$) and a significant increase in the p-type region for the Cu-containing Bi-Sb-Te (e.g., the 4Cu sample, $10.5 \pm 0.5 \times 10^{-4}$ W m$^{-1}$ K$^{-2}$), as shown in Fig. 3c. However, the precise trend in $PF$ changes for both p-type and n-type regions cannot be fully confirmed herein due to the limited data points, stemming from constraints in the feeding mechanism of our fabrication system. It should be noted that when the metal feed-rate exceeds 4, the film exhibits metallic behavior rather than that of a thermoelectric semiconductor, resulting in a negligible Seebeck coefficient. The 4Cu-Bi-Sb-Te alloy exhibits the best $PF$ with p-type characteristics. In this context, the Bi-Sb-Te becomes the minor phase and the metal-based phase might start to dominate, accounting for its metal-like $\rho$. In this case, a high $PF$ is achieved because the materials maintain a comparable $S$, suggesting the possible coexistence of other thermoelectric phases, such as $Cu_2Te$[59], with a high Cu content. Table S1 (Supplementary Information) summarizes sputtered or evaporated (R2R compatible deposition) Bi-Sb-Te in the recent two decades. These laboratory-scale studies typically focus on hot deposition or post-deposition annealing to enhance $PF$, but hot processing is not feasible in R2R manufacturing in our case. Compared to the Bi-Sb-Te film deposited under hot conditions (see Figure S3a, Supplementary Information), the pristine Bi-Sb-Te film in this study exhibits a moderate $PF$. This reduction in performance could be attributed to the poor crystallinity of our film deposited at room temperature, which is constrained by the R2R processing. The most comparable study (i.e. room temperature, non-annealed, similar film thickness, see Table S2, Supplementary Information) had a $PF$ of $0.9 \times 10^{-4}$ W m$^{-1}$ K$^{-2}$, which was approximately 10 times less than the highest $PF$ obtained here ($10.5 \pm 0.5 \times 10^{-4}$ W m$^{-1}$ K$^{-2}$). This value was comparable to some hot-deposition/annealed Bi-Sb-Te. Figure 3f compares Cu- or Ag-containing Bi-Sb-Te materials. The doped Bi-Sb-Te bulk has been extensively studied since 2005, achieving a $PF$ of ~$30 \times 10^{-4}$ W m$^{-1}$ K$^{-2}$. However, to the authors' knowledge, only three studies have explored the thin-film format of Cu- or Ag-doped Bi-Sb-Te, all focusing on hot processing during deposition or post-annealing. These studies reported a maximum $PF$ of ~$8 \times 10^{-4}$ W m$^{-1}$ K$^{-2}$, which is still lower than the highest $PF$ ($10.5 \pm 0.5 \times 10^{-4}$ W m$^{-1}$ K$^{-2}$ for the 4Cu-Bi-Sb-Te alloy) achieved here with room-temperature processed materials. Surprisingly, despite the high metal content in our films (as compared to the literature in Fig. 3f), they retain some degree of semiconducting (i.e., thermoelectric) properties. This can be attributed to our specific fabrication technique - alternating sputtering and evaporation of sub-nanometer-thick, non-continuous layers - which leads to a heterogeneous structure comprising a mixture of semiconductor phases, metal-rich phases, and pure metal inclusions. It should be noted that in comparison to all relevant works listed in Tables S1 & S3 (Supplementary Information), our work not only achieves excellent thermoelectric performance but also highlights a significant advancement in R2R large-scale manufacturing for large-area flexible materials. Most importantly, our technique demonstrates a unique patterning capability using SMT. Next, the parameters for the 4Cu-Bi-Sb-Te alloy will be utilized for the subsequent SMT study.

## 2.3 SMT-produced in-plane thermoelectric device

The oil used in SMT contains fluorine (F) and is generally highly reactive with deposited materials. In our previous study on SMT[4], we observed that residual oil could diffuse into the functional coating, leading to a non-continuous film and a loss of electrical performance after post-deposition cleaning. This issue was also observed here when sputtering of Bi-Sb-Te with SMT without co-evaporation of metals, resulting in non-conductive materials, even after post-cleaning with e-beam or plasma. However, we discovered that adding Cu through evaporation could suppress oil diffusion (Fig. 4a). To confirm the effect of Cu on oil diffusion, the spreading rate (increase in area) of an oil drop on the surface of the film was measured. As shown in Figure S4a, b) (Supplementary Information), the oil spreading rate slightly decreased in the Cu-Bi-Sb-Te, compared with the pure Bi-Sb-Te control. This, in turn, slowed down oil spreading when the droplet crossed a boundary from an uncoated to a coated region of the substrate (Figure S4c, d, Supplementary Information). Restricted oil spreading aided a more efficient cleaning process for the residual oil (Fig. 4a).

Four types of thermoelectric generator (TEG, see Table 2) were fabricated in a step-by-step process to replicate R2R fabrication: (1) Patterning of TE materials: replacing shadow mask (TEG 1) by SMT (TEG 2); (2) Patterning of metal contacts: replacing shadow mask (TEG 2) by printing metal contacts (TEG 3); (3) Cleaning residual oil post-SMT: replacing 1-h e-beam post-cleaning (TEG 3) by a few seconds plasma cleaning (TEG 4). The device resistance ($R_{TEG}$) of TEG with four pairs of thermoelectric strip/metal contact was measured using a two-probe method. $R_{TEG}$ increased from 4 kΩ (TEG 1) to 691 kΩ (TEG 2) due to the reaction between the film and F in the oil during SMT, indicating the formation of Cu-Bi-Sb-Te-F. TEG 1 represents the control group, consisting of shadow-mask patterned layers of the 4Cu-Bi-Sb-Te alloys discussed in previous sections. TEG 2 was fabricated using SMT combined with sputtering and evaporation, followed by e-beam post-cleaning. In this process, F elements from the oil used in SMT could be incorporated into the Cu-Bi-Sb-Te system, as previously confirmed in our work on n-type Bi-Te-F materials[4]. Herein it was observed that the SMT + sputtered Bi-Sb-Te, resulting in Bi-Sb-Te-F, was non-conductive. However, when the evaporated Cu source was added, the resulting Cu-Bi-Sb-Te-F material became conductive with p-type characteristics, achieving performance comparable to the control group (Cu-Bi-Sb-Te) (Fig. 4c).

In comparison with using SMT to make the metal contacts[4], employing inkjet printing of Ag for the contact made the process much easier for this prototype study. However, it significantly increased the $R_{TEG}$ to 2193 kΩ (TEG 2 vs. TEG 3). This increase could be attributed to high contact resistance at the interface between Cu-Bi-Sb-Te-F and the bulk Ag (e.g., void-induced) and/or the electrical resistance of bulk Ag being higher than that of thin-film Cu. Both e-beam and plasma methods were used successfully to clean residual oil. A surprising finding was that the plasma-cleaned device (TEG 4) was functional, indicating that this technique (SMT + plasma cleaning) could be viable for the manufacture of functional materials. A few seconds of plasma cleaning is compatible with a fast R2R process, and it is less aggressive than 1 h of e-beam cleaning ($R_{TEG}$: 1846 kΩ vs 2193 kΩ).

High-resistance flexible thin-film thermoelectrics are suitable for high-precision temperature or radiation detection. Here, we explore their potential application as power generators. The Seebeck coefficient of the device ($S_{TEG}$, for a device with four pairs of thermoelectric strips) can be determined from the slope of the open-circuit voltage plot (see Figure S3 b, Supplementary Information). The e-beam-cleaned sample ($S_{TEG\ 3}$) showed a coefficient of 0.15 mV/K, while the plasma-cleaned sample ($S_{TEG\ 4}$) exhibited a slightly higher coefficient

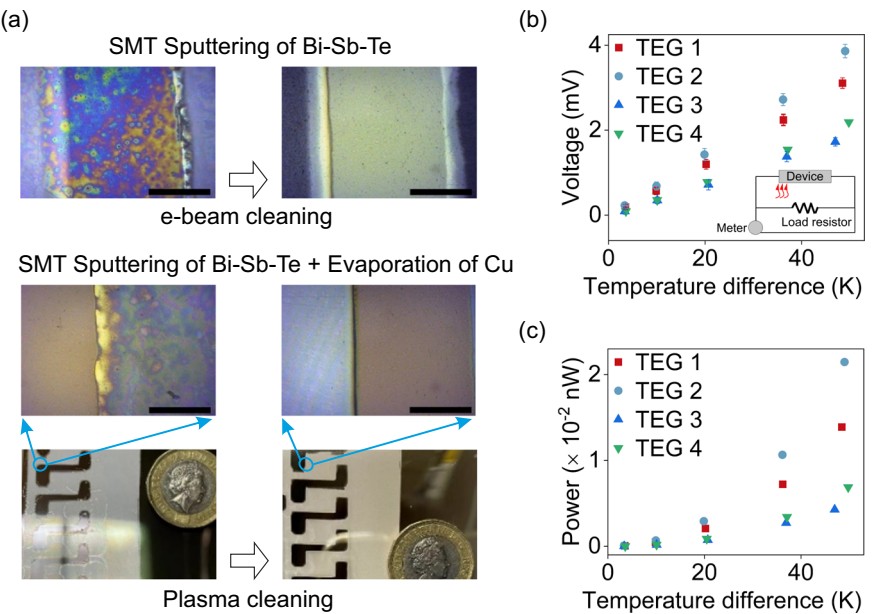

**Fig. 4 | Performance of SMT-Fabricated Thin-Film Thermoelectrics. a** Optical microscope images of SMT fabricated Bi-Sb-Te and Cu-Bi-Sb-Te before/after post-deposition cleaning (Scale bar = 1 mm). **b**, **c** The measured voltage and calculated power of TEGs (*Power = Voltage$^2$/$R_L$*). The device has four pairs of thermoelectric strips, and the load resistance ($R_L$) is 695 ± 1 kΩ.

**Table 2 | Fabrication details of TEGs (Sp. denotes sputtering and Evap. denotes evaporation)**

| Device | Thermoelectric patterns | Metal contact | $R_{TEG}$ (kΩ) | R2R compatible? |
|---|---|---|---|---|
| TEG 1 | Shadow Mask Sp. of Bi-Sb-Te + Evap. of Cu | Shadow Mask Evap. of Cu | 4 (± 0.1) | ☹ |
| TEG 2 | SMT Sp. of Bi-Sb-Te + Evap. of Cu, then 1 h e-beam cleaning (in situ) | Shadow Mask Evap. of Cu | 691 (±1) | ☹ |
| TEG 3 | Same to TEG 2 (1 h e-beam) | Ag printing | 2193 (±22) | 😐 |
| TEG 4 | SMT Sp. of Bi-Sb-Te + Evap. of Cu, then a few seconds of plasma (ex situ) | Ag printing | 1846 (±3) | 😊 |

$R_{TEG}$ is the internal resistance of the thermoelectric generator device. Shadow masks and 1 h e-beam processes are not compatible with R2R manufacturing, while SMT, a few seconds of plasma exposure and Ag printing are compatible with R2R.

of 0.16 mV/K. These values, along with $R_{TEG}$, explain the higher power output observed in the plasma-cleaned device compared to the e-beam-cleaned device, as shown in Fig. 4c. The extended 1-h e-beam exposure for the oil cleaning may have been too aggressive, potentially compromising the material's performance. However, as noted in our previous study[4], shorter e-beam cleaning times were insufficient to fully remove the residual oil. In contrast, plasma cleaning is rapid, taking only a few seconds, and is therefore more suitable for high-throughput R2R processes. This effectiveness is likely due to the interaction between the F-based oil and the plasma, where the active element F could play a significant role. The plasma used in this study was generated using the HiTUS technique (Plasma Quest Ltd., UK), which allows for remote plasma generation and precise direction using electromagnets, making it highly compatible with R2R processes.

## 2.4 Wristband design for wearable thin-film thermoelectrics
The SMT-produced planar TEGs were further processed into a wearable device (see Figure S5 & S6 for details, Supplementary Information). As illustrated in Fig. 5a, the in-plane TEGs were wrapped into a coil and then inserted into a 3D-printed wristband. PDMS molding was used to enhance the wristband's compatibility as a wearable device. To make the wristband thinner, the strip length for the SMT-fabricated in-plane TEGs needed to be as short as possible. Here, the optimized strip dimensions were predicted using COMSOL simulation (see Figure S7, Supplementary Information). The simulation results align well with the experimental data in ref. 60, and we explored an even smaller range for the strip size, which was not easily obtained from experimental

characterization. In Fig. 5b, the thermoelectric strip size was optimized to be approximately 3 mm in length and 6 mm in width (see Supplementary Information for COMSOL simulation analysis).

The fabricated TEGs were characterized, including an SMT-fabricated in-plane TEG, a cell TEG (25 pairs of in-plane TEGs), and a wearable wristband with 9 cells (see Figure S8, Supplementary Information). The TEG cell showed a much lower power output than the 3-pair in-plane TEG due to poor contact with the hot side. By adding PDMS to the 3D-printed frame, the thermoelectric performance was significantly improved. In this study, the prototype achieved an open-circuit voltage of approximately 90 mV during wearable application (see Fig. 5c). It should be noted that this voltage supply is generated by the temperature difference between the human skin and the ambient environment, making it a green, continuous, and reliable source. The voltage output of our device, which lies within the mV range as reported in ref. 61, can be integrated with a converter and is sufficient to power a wristwatch. Additionally, this mV voltage level is suitable for use in wearables and implantables requiring low-level voltage supply sources[62–67].

In addition to material improvements, we believe the device output can be further enhanced through optimizing the integration process. Figure S6e in the Supplementary Information shows the resistance changes recorded after each fabrication step. The winding process does not significantly affect device resistance, likely due to the protective PDMS layer on top, despite involving oven processing. It should be noted that the rolling direction is perpendicular to the direction of the thermoelectric strip, which may slightly reduce the

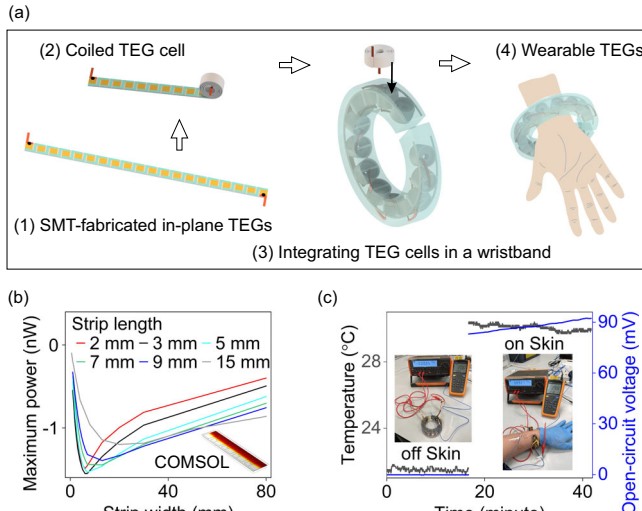

**Fig. 5 | Wearable Thermoelectric Wristband: Design, Optimization, and Performance. a** Schematic of device fabrication, from in-plane TEG to TEG cell, and finally to TEG wrist strap. **b** COMSOL simulation for optimizing thermoelectric strip size. **c** Thermoelectric power output of a wristband with 9 TEG cells. The inset image shows the measurement setup. The red cables connect the device to a multimeter for voltage recording, while the blue cable connects to a thermocouple for measuring skin temperature. The slight increase in the recorded voltage during the first 20 minutes in contact with the skin suggests the device is warming up, leading to a higher temperature difference and consequently a higher voltage output.

rolling-induced strain on the materials. However, integrating the device into the wrist frame increases resistance, highlighting the need for optimization at this step. Another significant resistance increase occurs during the connection of TEG cells in the frame, where Cu tape contacts are connected in series with Ag glue, and then cured in an oven (Step 6, Figure S5, Supplementary Information). The frame is subsequently embedded in PDMS and cured again (Step 7). These processes lead to a substantial rise in resistance to the MΩ range, which limits device output. Ideally, resistance should be within the kΩ range. For example, the resistance of a single thermoelectric pair in a flat condition is approximately 1 kΩ. Given that the prototype comprises 9 TEG cells, each with 25 thermoelectric pairs (totaling 225 pairs in the wrist strap), the ideal resistance would be around 225 kΩ, excluding contact resistance between TEG cells. Therefore, there is considerable potential to optimize the integration process, particularly the contacts between TEG cells. Additionally, the contact between the device and the heat source (i.e., human skin) should be improved. After PDMS molding (Step 7, Figure S5, Supplementary Information), direct contact between the TEG cells and the skin is prevented, limiting thermal contact and power output. Replacing the skin-side PDMS with a material of higher thermal conductivity could mitigate this issue. Furthermore, miniaturizing the TEG cells and optimizing the 3D-printed frame to integrate more cells into the strap could further enhance power output.

In conclusion, we investigated the integration of SMT with simultaneous sputtering and evaporation for large-scale patterning of flexible functional materials with nanometer thickness. High-throughput roll-to-roll processing was achieved using a 0.8 meter wide polymer web at an in-line speed of 25 m min⁻¹ to manufacture high-performance and patterned thin-film thermoelectrics. We combined the sputtering of Bi-Sb-Te thermoelectric materials with evaporation of Cu or Ag, to explore the effect of metal source in a semiconductor thin-film system. Significantly, this approach effectively addressed the deleterious effect of the SMT oil on the functionality of the deposited functional materials. Therefore, this roll-to-roll style SMT technique makes significant progress in patterning nano-thick functional materials in an industrial setting for large-scale manufacturing.

For continuous films fabricated using our roll-to-roll room-temperature deposition tool, the Cu- or Ag-containing Bi-Sb-Te-based composite materials exhibited an interesting n-type-to-p-type transition, influenced by the doping mechanism or the formation of other material phases. In the n-type region, Ag-Bi-Sb-Te demonstrated enhanced thermoelectric performance (power factor) from 0.01 to 1.0 ($\times 10^{-4}$ W m⁻¹ K⁻²). In the p-type region, Cu-Bi-Sb-Te achieved a remarkable maximum power factor of $10.5 \times 10^{-4}$ W m⁻¹ K⁻², comparable to Bi-Sb-Te-based materials fabricated and/or post-annealed under hot conditions. The SMT-fabricated in-plane thermoelectric device exhibits excellent thermoelectric performance and was designed into a coiled-up wristband for wearable applications.

## Methods
### Materials
The sputtering condition on a three-inch target ($Bi_{0.5}Sb_{1.5}Te_3$, 99.999%, Mi-Net Technology Ltd.) was fixed at 250 sccm Ar flow, 0.25 kW direct-current power, and 12 cm target-to-substrate distance. The evaporation was conducted under ~700 A current through a TiB2/BN ceramic boat (Sintec Ceramics) with a source-to-substrate distance of 180 mm.

To vary the content of the metal source (Cu or Ag), the feed-rate of metal wire (Ag: 1 mm diameter, 99.99%, Kurt J. Lesker Co. Ltd.; Cu: 1.2 mm diameter, Speedmet 98 Welding Alloys) was adjusted at number 1, 2, 3, 4 in the Aerre Machines coater. The number 20 corresponds to an in-line speed of 8 cm min⁻¹. It was checked that: < 1, no change in electrical performance; > 4, loss of Seebeck property i.e., metal behavior. The substrate was fixed to the coating drum which was rotated at an in-line speed of 25 m min⁻¹, the base pressure of the vacuum was ~1.3 × 10⁻⁴ mbar and the working pressure was ~6.7 × 10⁻⁴ mbar after the Ar flow and the evaporation sources were on. The samples are designated as nCu or nAg where n is the rate of the wire feed for evaporation.

### SMT for in-plane TEGs
SMT was conducted in an R2R web coater (Aerre Machines) using a 0.8 meter wide polymer web at an in-line speed of 25 m min⁻¹. A 36 μm thick polyethylene terephthalate (PET film, Mylar® A) was used as the flexible polymer substrate. The materials and deposition conditions have been described in the previous section. The post-cleaning process could be either an in situ e-beam (1 h, 7.74 V, 100 mA, 180 sccm Ar gas) or an ex situ plasma (10 s, 10 kW, 100 sccm Ar gas) - HiTUS technology (Plasma Quest Ltd.).

To electrically connect the thermoelectric patterns, SMT can be used to deposit metal interconnects, as previously demonstrated[4]. Herein, to simplify the fabrication process, a custom-built inkjet printing of Ag (DAG 1415 M) was used to connect the fabricated Cu- or Ag-doped Bi-Sb-Te patterns. Alternatively to the inkjet printing connected to TEG, a shadow mask was used as a control to deposit the contact materials via Cu evaporation, as described in the previous session.

### Wristband design for wearable applications
COMSOL simulation was used to guide the expected optimal length of the thermoelectric strip[68]. A p-n TEG was fabricated for wearable configuration. To simplify the fabrication process, p-type patterns were produced through SMT, while a shadow mask was employed to fabricate n-type Bi-Te patterns (using the deposition parameter as reported in previous studies[55,56], i.e., 250 sccm Ar gas, 0.25 kW direct-current power, 3-min sputtering).

As illustrated in Figure S5 (Supplementary Information), the fabricated p-n conjunct TEG was connected with Cu leads using silver glue at the end. Subsequently, a thin layer of PDMS was applied on top of the in-plane TEG through screen printing, utilizing a flexible mask (49-μm thick PET cut by TS 3040 40 W Laser Cutter). The 49-μm thick PDMS layer on

top aimed to enhance the device's flexibility[69] while facilitating the winding of the in-plane geometry into a roll. The rolls, representing the TEG cell, were injected into a 3D printed wristband (machine: Markforged M2; materials: Markforged Onyx Filament 800cc), as depicted in Figure S5 step 6 (Supplementary Information). Following the connection of these TEG cells by Cu leads using silver glue, the wristband was molded using PDMS to improve flexibility and durability.

## Materials/Device characterization

A DekTak stylus profilometer (Veeco 6 M) was used to measure the film thickness on a Si wafer, across a step height between the coating and the substrate partially masked by a polyimide tape before deposition (the results were averaged from ten different locations).

The film phase was identified using XRD (Rigaku Miniflex diffractometer): Cu $k_\alpha$ radiation (0.154 nm), 40 mA, 40 kV, 10°–80° (2θ) and 0.007° step size. The characterized film was deposited on a mechanical-grade (1196) silicon wafer.

The film surface morphology was imaged in µm size by field emission-SEM, Zeiss Merlin. The SEM condition was 100 kX of magnification, 3 kV of voltage, 100 pA of the current probe, and 6.6 mm of working distance. The film composition was analyzed using EDX integrated with SEM, using a point ID mode over eight independent locations, under 10.5 mm working distance, 5 kV voltage, and 2.5 kX magnification.

The atomic size of the film surface morphology was captured by AFM (Agilent 5400) in tapping mode using an NCHV-A tip (Bruker Ltd.). The image was processed using WSxM 5.0 Develop 9.0 software.

The sheet resistance of the film was obtained from an in-house custom four-point probe system at nine locations, by applying current in the outer two probes and measuring voltage in two inner probes using an Agilent 34420 A meter. The electrical resistivity was then calculated using the following equation:

$$\rho = R_s \cdot t = \frac{\pi}{\ln 2} \cdot \frac{V}{I} \cdot t \qquad (2)$$

where $\rho$ is the electrical resistivity, $R_s$ is the sheet resistance, $V$ is the voltage measured, $I$ is the current applied, and $t$ is the film thickness.

The carrier concentration and mobility of the thin film were measured in a home-built Hall measurement system at room temperature (magnet: 0.165 T) with a van der Pauw configuration using Keysight Technologies B2901A. Three identical specimens (1 cm × 1 cm) with Cu contacts at the four corners (0.1 cm × 0.1 cm) were measured for the mean.

The oil dispersion on the film was analyzed at room temperature and atmospheric pressure. 10 µL of the oil was dropped on top of the film, then images were captured via Dino-Lite digital microscope at 20 – 60 s. The dispersion rate was characterized by plotting the size of a droplet (measured using ImageJ) with time.

The sample was imaged using a phone camera and optical microscope (Yenway Microscopes).

The Seebeck coefficient of the thin film was measured using a Seebeck system (MMR Technologies Inc.) under a temperature difference of 0.1–0.45 K, at room temperature in a nitrogen atmosphere. Seebeck coefficient was calculated by the ratio of the output voltage to the temperature difference applied. An average result was obtained from at least nine measurements.

Seebeck performance of TEG was assessed using a home built Seebeck setup using two Peltier modules (European Thermodynamics Ltd.) to apply a temperature difference and the voltage output was measured in a simple Ohm's circuit, where the load resistance was same to the resistance of the device to get the maximum power output[60]. By measuring the voltage output ($V$) across a load resistor ($R_L$), the power was calculated by $V^2/R_L$.

The wrist-strap design required the length of thermoelectric strips to be short to minimize the thickness of the wristband. COMSOL simulation[68] was used to predict the optimized geometry of the thermoelectric strip i.e. the length and the width. In Figure S7 (Supplementary Information), the simulated results closely align with the experimental data reported in[60]. Additionally, a predicted range is provided to minimize the strip size, which may not be readily obtained through experimental measurements.

The stability of the material's resistivity was characterized using a custom in-house four-point probe system at 15 different locations across three identical samples. These samples were taken from three different locations on the coating drum (radius: 0.3 m, width: 0.8 m), randomly selected from a total sample size of 1.8 m × 15 cm. Although the coating drum has a width of 0.8 m, we selected samples from the middle region (15 cm wide) due to the comparatively small diameter of the sputtering target. As shown in Figure S9 (Supplementary Information), the material exhibited negligible resistance change over two weeks of storage under ambient conditions. Additionally, the error bars indicate uniformity across the coating drum. Batch-to-batch variation was assessed through three identical material fabrication batches, revealing a 3% variation in Seebeck characteristics. For the device, the thermoelectric pair exhibited a resistance variation of approximately 6%. These results confirm the uniformity of the films fabricated using our large-scale manufacturing technique.

## Reporting summary

Further information on research design is available in the Nature Portfolio Reporting Summary linked to this article.

## Data availability

The data generated in this study are provided in the Supplementary Source Data file. Source data are provided with this paper.

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

## Acknowledgements

The authors would like to acknowledge funding from the Engineering and Physical Sciences Research Council [grant number EP/M015173/1] and the EPSRC Impact Acceleration Account Award [EP/R511742/1] (H. E. A. and X.T.). R.S.B was supported by the Royal Academy of Engineering under the Research Fellowship scheme (RF\201819\18\38). E. B. would like to thank the support from UKRI Innovate UK (KiriTEG Project, Reference: 51868). The authors are also grateful to the equipment access from the Oxford Materials Characterisation Service and the David Cockayne Centre for Electron Microscopy, with the Department of Materials Oxford University. We would like to acknowledge the help of Steve J. Wakeham, James D. Dutson, and Mike J. Thwaites from Plasma Quest Ltd. UK. We greatly acknowledge Bryan W. Stuart for the video recording and technical support.

## Author contributions

X.T. designed the project, conducted the experiments, analyzed the data, wrote the initial draft, and revised the manuscript. Q.Z., C.Z., H.P., Z.Z., and J.E. performed experiments. R.S.B., E.B., and P.S.G. contributed to the manuscript revision and provided technical support. H.E.A. supervised the project, secured the funding, and contributed to the manuscript revision.

## Competing interests

The authors declare no competing interests.
