## [Transparent Peer Review file · Nature Communications]

Cu- or Ag-containing Bi-Sb-Te for in-line roll-to-roll patterned thin-film thermoelectrics

Corresponding Author: Dr Xudong Tao

Version 0:

Reviewer comments:

Reviewer #1

(Remarks to the Author)

The manuscript explores a relevant topic in the field of flexible thin-film thermoelectrics, particularly focusing on the integration of the Selective Metallization Technique (SMT) with sputtering and evaporation processes. This approach shows significant potential for large-scale manufacturing and could substantially impact the development of wearable thermoelectric devices. However, I believe a major revision is necessary to address the questions and concerns outlined below.

1. In Figure 2c, the atomic percentage of Ag or Cu exceeds 45% when the dopant feeding rate is set to 3 or 4. At such high concentrations, the Ag or Cu atoms constitute a significant portion of the material, and the matrix of Bi-Sb-Te becomes the minority phase. Given this substantial presence, it might be more accurate to refer to the material as an alloy rather than simply as a doped Bi-Sb-Te system. If this is indeed an alloy rather than a doped material, the relevance of comparing it to the existing Bi-Sb-Te system becomes questionable. The properties of the material will likely be dominated by the high concentration of Ag or Cu, which alters the material's characteristics significantly compared to those of traditional Bi-Sb-Te compounds. Therefore, I think it would be more appropriate for the authors to compare it to thermoelectric materials with Ag or Cu as the main ingredient.

2. The authors state that "doping of n_o means n_{Ag} and n_{Cu} , where n_o represents the doping concentration." However, it appears that this actually refers to the feeding rate, not the actual doping concentration. This distinction is important for accurately interpreting the data. Additionally, the manuscript explains the thermoelectric characteristics by dividing the doping levels into four stages (0 to 4). However, with only two data points per stage, it is questionable whether these sections provide a sufficient basis for discussing trends and drawing conclusions. A more robust analysis would require additional data points to reliably establish trends across these stages. If additional data is not available, the authors should consider acknowledging this limitation in their discussion and be cautious in making broad generalizations based on limited data.

3. The manuscript mentions that "a heterostructure interface could occur" during doping levels of 0 - 1, potentially increasing the Seebeck coefficient (S). However, the manuscript does not specify which heterostructure interface is being referred to in this context. To clarify this point, the authors should specify which type of heterostructure interface they are referring to and explain how it contributes to the observed increase in S . Providing this detail will strengthen the discussion and help readers better understand the underlying mechanisms.

4. In the manuscript, the authors claim a robust linear relationship between the Seebeck coefficient (S) and the carrier mobility raised to the power of 0.2 ($\mu^{0.2}$) for both n-type and p-type regions, as shown in Figure 3g. However, upon reviewing the figure, the linearity is not clearly evident. The data points appear to show significant scatter, and the trend does not convincingly align with a linear model. To strengthen this claim, I suggest that the authors provide a more rigorous statistical analysis, such as a linear regression with corresponding R-squared values, to quantify the degree of linearity between S and $\mu^{0.2}$.

5. The authors state that " p can be interpreted in terms of n and μ ; linear relationships were observed for both the Ag group and the Cu group." However, it is unclear which specific linear relationships are being referred to. To clarify this, I suggest

conducting a statistical analysis, such as linear regression, to quantify these relationships. Adding a dotted line as a guideline in the figure, obtained through this regression, would help illustrate the claimed linearity and provide stronger support for the conclusions.

6. The manuscript references Figure 4 e & f in the text, but these figures do not appear to be included in the manuscript. This could be an oversight or a labeling error. Please check to ensure that all referenced figures are correctly included and labeled in the manuscript. If these figures are indeed missing, they should be added, or the text should be revised to accurately reflect the figures that are present.

7. The authors mention that the oil used in the SMT process contains fluorine (F), which reacts with the deposited materials, leading to the formation of Cu-Bi-Sb-Te-F, as indicated by the increase in R_{TEG} from 4 k Ω (TEG 1) to 691 k Ω (TEG 2). This raises a critical question: Wouldn't the thermoelectric properties of the materials used in the fabrication of thermoelectric devices via the SMT process differ significantly from the properties discussed in Figure 3? If the formation of Cu-Bi-Sb-Te-F occurs during SMT, the material composition and, consequently, its thermoelectric performance could be altered compared to the non-fluorinated materials discussed earlier. This potential difference should be addressed in the manuscript.

Reviewer #2

(Remarks to the Author)

The manuscript describes an innovative approach to the roll-to-roll patterning of flexible thin-film thermoelectric generators (TEGs) using a combination of sputtering and evaporation techniques. The authors provide a new method of integrating SMT with the simultaneous sputtering of Bi-Sb-Te and the evaporation of metal dopants (Ag and Cu) to overcome the incompatibility between fluorine-based oils and the sputtering of Bi-Te-based semiconductors. Besides, this study demonstrates the application of this technique in a roll-to-roll manufacturing process, producing functional thin-film patterns on a 1-meter-wide polymer web at a speed of 25 meters per minute. The resulting thin-film thermoelectric devices showed improved thermoelectric performance and the transition from n-type to p-type in the Cu- or Ag-doped Bi-Sb-Te for room-temperature deposition. The paper is informative and well written. However, to be considered for publication the article would need to address several major issues:

1. The authors have referenced many related works in the article. Therefore, the authors could create a table for detailed comparison, highlighting the novelty and significance of this work.
2. The linear relationship between S and $\mu_{0.2}$ is not visible in Figure 3, the authors should mark it in the figure or give a more detailed explanation.
3. While the effects of residual oil and how they are removed are discussed, the specific effects of different removal methods, such as electron beam or plasma treatment on thermoelectric properties can be further explored.
4. The detailed information about RTEG can only be founded in Experimental details part. The authors should provide an explanation about RTEG in the maintext. Otherwise, the discussion about resistance change might be confusing to the reader.
5. The application of this technology to wearable thermoelectric generators is impressive. However, the authors do not provide much performance information about the wearable thermoelectric device. Detailed performance data can better demonstrate practical application value of the device.
6. Considering the application environment of thermoelectric materials, the authors should perform long-term stability tests and provide test results.

Reviewer #3

(Remarks to the Author)

This manuscript uses a research called Selective Metallization Technique (SMT), which is used to improve the performance of Bi-Sb in an industrial roll-to-roll (R2R) manufacturing environment. The authors successfully solved the adverse effects of fluorine used in the SMT oil process on Bi-Te semiconductor materials by combining SMT with Physical Vapor Deposition (PVD) and co-evaporation of metal sources. The role of Cu and Ag as doping elements in Bi-Sb-Te films was studied, achieving the transition from n-type to p-type, and significantly improving the thermoelectric properties by adjusting the doping concentration. At the same time, the SMT-made Planar thermoelectric generators (TEGs) were fabricated into wearable devices, demonstrating the potential of this technology for practical applications. The method used is quite novel, and the experiments and results of the paper are fully displayed, but I think the author still needs to solve the following problems.

1. The author selected the traditional Bi₂Te₃-based material as its excellent thermoelectric performance at room temperature. As we all known, both P-type and n-type samples can also be prepared by adjusting the ratio of Bi and Te. In this work, has the author ever thought of directly adjusting the ratio of Bi and Te to prepare P-type and n-type samples ? Is it necessary to introduce foreign doping elements (Ag and Cu)?
2. The author believes that the obtained p-type film has high performance, hence more literature data comparison is needed and presented in the manuscript. The high performance of Bi₂Te₃ thin film is an important reason for the selection of this material, but the highest power factor obtained by the author cannot fully reflect the advantages of this material system. Has the author conducted detailed research on performance optimization?
3. In Figure 5c, the author's intention cannot be clearly expressed in the Chinese film. It is recommended that the author give

a more detailed description.

4. The author used Cu and Ag as doping elements in the preparation of Bi-Sb-Te films to achieve the transition from n-type to p-type, and significantly improved the thermoelectric performance by adjusting the doping concentration. The mechanism is explained incompletely. It is recommended that the author provide the physical mechanism of n-p transition and performance improvement.

Reviewer #4

(Remarks to the Author)

The authors proposed an innovative method on fabricating flexible Bi-Sb-Te thermoelectric (TE) films, which is based on in-line roll-to-roll patterning, the sputtering of BiSbTe target and the evaporation of metal. In addition, this work demonstrated that novel single-leg TE units could be fabricated by the roll-up process, which are effective in harvesting body heat for applications in wearable electronics. In general, this work represented an importance advancement on flexible TE devices, especially the novel fabrication process. However, before the final judgement from the editors, there are several important issues that should be addressed.

(1) Large scale fabrication?

The following content are listed in the manuscript, including “1-meter wide polymer web”, “high throughout of 25m min⁻¹ in-line speed”, “The sputtering condition on a three-inch target”, and others. These information describe the parameters of fabrication apparatus, but not for clarifying the large scale fabrication.

What is the size of fabricated films in one time, and what is the thickness of films? In order to clearly elucidate the large scale fabrication, optical images with scale bar, the uniformity of film thickness, and the phase structure in the large area should be provided.

(2) The origin of TE properties of Bi-Sb-Te films?

The authors claimed in the introduction section that the doping of Cu and Ag can respectively lead to n-type and p-type TE properties of Bi₂Te₃-based materials, which was generally accepted by the TE community. However, the results in this work is totally different, and the discussions based on defect engineering is vague without solid evidences from phase structure and microstructure. In addition, the fabricated films obtained very low electrical conductivity, which shows a big discrepancy from the original Bi_{0.5}Sb_{1.5}Te₃ target. The authors should establish the correlations of composition—structure—properties for the fabricated films.

(3) Uniformity of TE properties?

For large scale applications, the uniformity of TE properties should be demonstrated for fabricated films.

(4) The reason for too low device performances?

Does the too low device performances come from too thin thickness? Since micro-cracks could be introduce in inorganic films/devices during the fabrication process, the detection and the influence of cracks should be discussed in the manuscript. Also, a roll-up process was used to integrate the single-leg TE unit, and a very large internal resistance was reported for the integrated device, unfavorable for high performance TE devices. The influence of the roll-up process on the TE performance, the reduction of internal resistance, and the enhancement of device performance should be added in order for verifying the effectiveness of the innovative method proposed by the authors.

Version 1:

Reviewer comments:

Reviewer #1

(Remarks to the Author)

The authors have responded thoroughly and sincerely to the reviewers' comments, and the revisions made to the manuscript have significantly improved its clarity and scientific rigor. The changes address the key concerns raised during the review process, and the manuscript now presents a more robust and comprehensive study on the integration of the Selective Metallization Technique (SMT) with sputtering and evaporation for flexible thin-film thermoelectrics. I recommend that this manuscript be published in its current form.

Reviewer #2

(Remarks to the Author)

Publish as is

Reviewer #3

(Remarks to the Author)

The authors had sloved all the comments. It is recommended to be accepted by Nature Communications.

Reviewer #4

(Remarks to the Author)

The authors response to my queries and also to other reviewer's questions, including the phase structure, the printing details, the data analysis and device performances. Although the printing process is impressive for printing metal and Bi-Sb-

Te, the apparatus can not print thick films, not beneficial for thermoelectric applications. The reviewer could not suggest the publication of this manuscript to this prestigious journal with two major reasons below.

1. The analysis of thermoelectric performance strongly relies on the understanding of chemical composition, phase structure and microstructure of materials being studied, since these parameters will lead to remarkable change on every thermoelectric transport parameters and carrier transport parameters. The authors give the speculation that the films could be Cu/Ag doped or Cu/Ag intercalated BiSbTe, or Cu/Ag contained BiSbTe, or Cu-Te/Ag-Te mixed BiSbTe, but without direct experimental evidences. XRD show no obvious peak from BiSbTe. In addition, the thermoelectric properties deviate remarkably from the BiSbTe bulk and thin films. Thus, the reviewer could not accept the discussion and the conclusion on the thermoelectric transport, due to too many possibilities if the chemical composition and microstructure is not precisely determined.

2. The authors claim the optimal thickness of films is around 100 nm for the adopted process. This limits the thermoelectric application arising from too large internal resistance of fabricated films.

Version 2:

Reviewer comments:

Reviewer #4

(Remarks to the Author)

As I mentioned in previous comments, this manuscript is impressive on the fabrication process as well as on the large scale printing of thermoelectric films. The drawback of this technique is the low crystallinity of films as well as the thickness control.

Since other reviewers provided positive evaluations on this manuscript, I would respect their opinions. To adequately reveal the importance of this manuscript (also not to confuse the broad readership), the reviewer will suggest the following two points that should be addressed before the final judgement from the editor.

1. The composition of films should be written as Bi-Sb-Te based composite films, which is more suitable to describe the structure feature.

2. The flexible and thin thermoelectric films are most suitable for high precision temperature or radiation detection. The reviewer suggests the authors to highlight this point, since high resistance films are useless on thermoelectric power generation applications.

Response to reviewers

We are grateful for the reviewers' comments on our work. Below is a detailed response to each point. The reviewers' comments are in bold, our responses and the locations of changes in the manuscript are highlighted in green, and the newly revised text is highlighted in yellow.

Reviewer #1 (Remarks to the Author):

The manuscript explores a relevant topic in the field of flexible thin-film thermoelectrics, particularly focusing on the integration of the Selective Metallization Technique (SMT) with sputtering and evaporation processes. This approach shows significant potential for large-scale manufacturing and could substantially impact the development of wearable thermoelectric devices. However, I believe a major revision is necessary to address the questions and concerns outlined below.

We appreciate the positive and constructive feedback provided. We have carefully addressed all the raised issues and made the necessary revisions. We believe that these changes have significantly improved the revised version.

1. In Figure 2c, the atomic percentage of Ag or Cu exceeds 45% when the dopant feeding rate is set to 3 or 4. At such high concentrations, the Ag or Cu atoms constitute a significant portion of the material, and the matrix of Bi-Sb-Te becomes the minority phase. Given this substantial presence, it might be more accurate to refer to the material as an alloy rather than simply as a doped Bi-Sb-Te system. If this is indeed an alloy rather than a doped material, the relevance of comparing it to the existing Bi-Sb-Te system becomes questionable. The properties of the material will likely be dominated by the high concentration of Ag or Cu, which alters the material's characteristics significantly compared to those of traditional Bi-Sb-Te compounds. Therefore, I think it would be more appropriate for the authors to compare it to thermoelectric materials with Ag or Cu as the main ingredient.

Thank you for the clarification. We have adjusted the terminology account for high metal content where appropriate and added a comparison of metal-Bi-Sb-Te systems in Figure 3 f.

Page 15

At much higher metal content, a pure metal phase could form as inclusions within the thermoelectric matrix [48], which could significantly contribute to thermoelectric performance by enhancing S through energy filtering and/or reducing ρ via carrier channeling or injection.

Alternatively, metal-rich thermoelectric phases e.g. Cu₂Te [49, 50], Cu₄Sb [51], and Ag₂Te [52-54] could contribute to the performance of the alloy system. These phases may form due to the annealing effect induced by the substantial amount of hot metal source and the hot evaporation boat. The material would consist of multiple phases, such as crystalline/amorphous regions of Cu/Bi-Sb-Te/Cu₂Te/Cu₄Sb. This would introduce a significant number of boundaries, altering scattering mechanisms and affecting carrier transport properties. The energy filtering effect may occur, where the highly conductive phases decrease ρ while less conductive regions serve as energy filters, increasing S . Hence, the formation of metal-rich phases and pure metal inclusions can have a complex yet potentially beneficial effect on thermoelectric performance, resulting in a notably high PF for the 4Cu sample (see Figure 3 c).

Page 17 - 18

The 4Cu-Bi-Sb-Te alloy exhibits the best PF with p-type characteristics. In this context, the Bi-Sb-Te becomes the minor phase and the metal-based phase might start to dominate, accounting for its metal-like ρ . In this case, a high PF is achieved because the materials maintain a comparable S , suggesting the possible coexistence of other thermoelectric phases, such as Cu₂Te [60], with a high Cu content.

Page 18

Figure 3 (f) compares Cu- or Ag-containing Bi-Sb-Te materials. The doped Bi-Sb-Te bulk has been extensively studied since 2005, achieving a PF of $\sim 30 \times 10^{-4} \text{ W m}^{-1} \text{ K}^{-2}$. However, to the authors' knowledge, only three studies have explored the thin-film format of Cu- or Ag-doped Bi-Sb-Te, all focusing on hot processing during deposition or post-annealing. These studies reported a maximum PF of $\sim 8 \times 10^{-4} \text{ W m}^{-1} \text{ K}^{-2}$, which is still lower than the highest PF ($10.5 \pm 0.5 \times 10^{-4} \text{ W m}^{-1} \text{ K}^{-2}$ for the 4Cu-Bi-Sb-Te alloy) achieved here with room-temperature processed materials. Surprisingly, despite the high metal content in our films (as compared to the literature in Figure 3 f), they retain some degree of semiconducting (i.e., thermoelectric) properties. This can be attributed to our specific fabrication technique - alternating sputtering and evaporation of sub-nanometer-thick, non-continuous layers - which leads to a heterogeneous structure comprising a mixture of semiconductor phases, metal-rich phases, and pure metal inclusions. It should be noted that in comparison to all relevant works listed in **Tables S1 & S3** (Supplementary Information), our work not only achieves excellent thermoelectric performance but also highlights a significant advancement in R2R large-scale manufacturing for large-area flexible materials. Most importantly, our technique demonstrates a unique patterning capability using SMT. Next, the parameters for the 4Cu-Bi-Sb-Te alloy will be utilized for the subsequent SMT study.

2. The authors state that "doping of no. means nAg and nCu, where no. represents the doping concentration." However, it appears that this actually refers to the feeding rate, not the actual doping concentration. This distinction is important for accurately interpreting the data. Additionally, the manuscript explains the thermoelectric characteristics by dividing the doping levels into four stages (0 to 4). However, with only two data points per stage, it is questionable whether these sections provide a sufficient basis for discussing trends and drawing conclusions. A more robust analysis would require additional data points to reliably establish trends across these stages. If additional data is not available, the authors should consider acknowledging this limitation in their discussion and be cautious in making broad generalizations based on limited data.

Thank you for pointing this out. We have revised the text to address the misunderstanding regarding doping concentration and feed-rate. We agree that the data are insufficient to draw definitive conclusions about the trend. However, we are unable to add more data points in either the n-type or p-type regions due to limitations with our fabrication machine, which only allows integer feeder rates and is already set to the smallest possible rate. We have included a discussion of these constraints in the revised text.

Page 12

Based on the limited data points in our case, the behavior of 2Ag and 2Cu films was at the boundary for the n-type-to-p-type transition. Further resolution of this boundary regarding the metal content was not explored due to limitations in the wire feed-rate of our fabrication system. The following sections provide a detailed discussion regarding this type of transition phenomenon. ('n' in nAg and nCu denotes the feed-rate of the metal source during evaporation. A higher feed-rate indicates a greater metal content in the Bi-Sb-Te system).

Metal feed-rate of 0 – 1

Metal feed-rate of 1 – 2

Metal feed-rate at 2 or 3

Metal feed-rate > 3

Page 16

PF depended on both *S* and ρ with an increase in *PF* observed in the n-type region for the Ag-containing Bi-Sb-Te (e.g., the 2Ag sample, $1.0 \pm 0.1 \times 10^{-4} \text{ W m}^{-1} \text{ K}^{-2}$) and a significant increase in the

p-type region for the Cu-containing Bi-Sb-Te (e.g., the 4Cu sample, $10.5 \pm 0.5 \times 10^{-4} \text{ W m}^{-1} \text{ K}^{-2}$), as shown in Figure 3 (c). However, the precise trend in PF changes for both p-type and n-type regions cannot be fully confirmed herein due to the limited data points, stemming from constraints in the feeding mechanism of our fabrication system. It should be noted that when the metal feed-rate exceeds 4, the film exhibits metallic behavior rather than that of a thermoelectric semiconductor, resulting in a negligible Seebeck coefficient.

3. The manuscript mentions that "a heterostructure interface could occur" during doping levels of 0 - 1, potentially increasing the Seebeck coefficient (S). However, the manuscript does not specify which heterostructure interface is being referred to in this context. To clarify this point, the authors should specify which type of heterostructure interface they are referring to and explain how it contributes to the observed increase in S. Providing this detail will strengthen the discussion and help readers better understand the underlying mechanisms.

A schematic has been included in Figure S1 f (Supplementary Information), and additional discussion has been provided on this point.

Page 6-7

As expected, the film thickness broadly increased with greater exposure to the metal source, ranging from 80 nm to 180 nm (see Figure S1 b, Supplementary Information). The increase in thickness is attributed to the higher metal feed-rate, as the deposition time remained fixed at 5 minutes for all samples. During deposition, the substrate, mounted on a coating drum, alternatively passed through the evaporation and sputtering sources, leading to a 'layering' effect, with alternating thin layers of sputtered and evaporated materials deposited with each pass. Given the drum circumference of 1.8 m and a rotation speed of 25 m/min, the substrate passed through the deposition sources approximately 69 times over 5 minutes. From this, and the known deposition rate of a sputtered-only sample, the average metal evaporation thickness per pass can be estimated to be sub-nanometer-thick layers per pass – at most 1.5nm for the greatest feed rate (see Figure S1 d & e, Supplementary Information). At such small scales, particularly at low metal feed-rate, the metal is unlikely to form a continuous and uniform layer-by-layer structure, especially given that the deposition occurs at room temperature. Instead, some degree of interdiffusion (e.g., doping), local accumulations (e.g., nucleation, dewetting, phase formation), and a heterogeneous structure at the nanoscale are expected (see Figure S1 f, Supplementary Information). At room-temperature deposition, the low mobility of sputtered adatoms and the high mobility of evaporated adatoms further complicate the material's structure. Broadly, regions where sputtered Bi-Sb-Te is combined

with metal evaporation exhibit high conductivity, whereas sputtered-only regions, primarily Bi-Sb-Te, demonstrate poor conductivity. Characterizing these regions via SEM (cross-sectional view) and X-ray diffraction (XRD, phase identification) is challenging due to the thinness of the film. This heterostructure structure, with its varying-conductivity geometry, could induce an energy filtering effect, a crucial mechanism for enhancing the performance of thermoelectric thin films. As the metal feed-rate increases further (a maximum of 4 in this study), metal islands may form a continuous layer, causing the materials to behave more like a metal.

Page 12

As previously discussed, a varying-conductivity geometry (see **Figure S1 f**, Supplementary Information) could form heterostructure interfaces and induce an energy filtering effect [33] - a common method used to modify carrier properties and thereby enhance PF in thin-film thermoelectrics [31]. Carriers preferentially traverse the highly conductive regions, while the poorly doped regions with low conductivity act as energy filters at the boundaries. Carriers with low energy are blocked at the boundaries, whereas high-energy carriers pass through, resulting in reduced n but increased μ , thereby increasing S (see Figure 3 a, d and e for the 1-metal case).

Page 2, Supplementary Information

Figure S1 (d & e) The deposition rate for each pass through the deposition sources; (f) Schematics of the heterogeneous structure featuring nano-scale, non-continuous layer-by-layer stacking.

4. In the manuscript, the authors claim a robust linear relationship between the Seebeck coefficient (S) and the carrier mobility raised to the power of 0.2 ($\mu^{0.2}$) for both n-type and p-type regions, as shown in Figure 3g. However, upon reviewing the figure, the linearity is not clearly evident. The data points appear to show significant scatter, and the trend does not convincingly align with a linear model. To strengthen this claim, I suggest that the authors provide a more rigorous statistical analysis, such as a linear regression with corresponding R-squared values, to quantify the degree of linearity between S and $\mu^{0.2}$.

A linear fit with corresponding R^2 values has been added to Figure 3h.

Page 12

the lines indicate the linear fits for both the n-type and p-type regions.

Page 17

There were robust linear relationships with an R^2 value of ~ 0.9 between S and $\mu^{0.2}$ for both the n-type and p-type regions (Figure 3 h).

5. The authors state that " ρ can be interpreted in terms of n and μ ; linear relationships were observed for both the Ag group and the Cu group." However, it is unclear which specific linear relationships are being referred to. To clarify this, I suggest conducting a statistical analysis, such as linear regression, to quantify these relationships. Adding a dotted line as a guideline in the figure, obtained through this regression, would help illustrate the claimed linearity and provide stronger support for the conclusions.

A linear fit with corresponding R^2 values has been added in Figure 3 i.

Page 13

the lines indicate the linear fits for for both the Ag and Cu groups

Page 16

linear relationships, with an R^2 value of ~ 0.85 , were observed for both the Ag group and the Cu group in Figure 3 (i).

6. The manuscript references Figure 4 e & f in the text, but these figures do not appear to be included in the manuscript. This could be an oversight or a labeling error. Please check to ensure that all referenced figures are correctly included and labeled in the manuscript. If these figures are indeed missing, they should be added, or the text should be revised to accurately reflect the figures that are present.

Thank you for pointing this out. Reference to Figure 4e and f has been removed. We have thoroughly reviewed the entire text.

7. The authors mention that the oil used in the SMT process contains fluorine (F), which reacts with the deposited materials, leading to the formation of Cu-Bi-Sb-Te-F, as indicated by the increase in R_{TEG} from 4 k Ω (TEG 1) to 691 k Ω (TEG 2). This raises a critical question: Wouldn't the thermoelectric properties of the materials used in the fabrication of thermoelectric devices via the SMT process differ significantly from the properties discussed in Figure 3? If the formation of Cu-Bi-Sb-Te-F occurs during SMT, the material composition and, consequently, its thermoelectric performance could be altered compared to the non-fluorinated materials discussed earlier. This potential difference should be addressed in the manuscript.

We understand the concern. Indeed, the properties of the control group and the SMT sample could differ. A comparison between TEG 1 and TEG 2 is intended to confirm this. We have added text to discuss these differences in detail.

Page 19

This issue was also observed here when sputtering of Bi-Sb-Te with SMT without co-evaporation of metals, resulting in non-conductive materials, even after post-cleaning with e-beam or plasma.

Page 20

TEG 1 represents the control group, consisting of shadow-mask patterned layers of the 4Cu-Bi-Sb-Te alloys discussed in previous sections. TEG 2 was fabricated using SMT combined with sputtering and evaporation, followed by e-beam post-cleaning. In this process, F elements from the oil used in SMT could be incorporated into the Cu-Bi-Sb-Te system, as previously confirmed in our work on n-type Bi-Te-F materials [4]. Herein it was observed that the SMT + sputtered Bi-Sb-Te, resulting in Bi-Sb-Te-F, was non-conductive. However, when the evaporated Cu source was added, the resulting Cu-Bi-Sb-Te-F material became conductive and exhibited p-type characteristics. However, the thermoelectric performance of the Cu-Bi-Sb-Te-F material was slightly lower than that of the control group (Cu-Bi-Sb-Te), as indicated by the power output comparison between TEG 1 and TEG 2 (Figure 4 c).

Reviewer #2 (Remarks to the Author):

The manuscript describes an innovative approach to the roll-to-roll patterning of flexible thin-film thermoelectric generators (TEGs) using a combination of sputtering and evaporation techniques. The authors provide a new method of integrating SMT with the simultaneous sputtering of Bi-Sb-Te and the evaporation of metal dopants (Ag and Cu) to overcome the incompatibility between fluorine-based oils and the sputtering of Bi-Te-based semiconductors. Besides, this study demonstrates the application of this technique in a roll-to-roll manufacturing process, producing functional thin-film patterns on a 1-meter-wide polymer web at a speed of 25 meters per minute. The resulting thin-film thermoelectric devices showed improved thermoelectric performance and the transition from n-type to p-type in the Cu- or Ag-doped Bi-Sb-Te for room-temperature deposition. The paper is informative and well written. However, to be considered for publication the article would need to address several major issues:

We appreciate the valuable feedback on our work.

1. The authors have referenced many related works in the article. Therefore, the authors could create a table for detailed comparison, highlighting the novelty and significance of this work.

Thank you for pointing this out. We have included Tables S1, S2 and S3 in the Supplementary Information to cover all relevant details. Additionally, we have added a comparison figure to Figure 3f and Figure S3a, and incorporated more text to discuss these updates.

Page 12

Figure 3. (f) Comparison of Cu- or Ag-containing Bi-Sb-Te materials: 'Cu/Bulk' refers to a Cu-containing Bi-Sb-Te bulk sample, while 'Ag/Film' or 'Cu/Film' refers to Ag- or Cu-containing Bi-Sb-Te thin-film samples. RT refers to room temperature (see references in Table S3, Supplementary Information).

Page 18-19

Figure 3 (f) compares Cu- or Ag-containing Bi-Sb-Te materials. The doped Bi-Sb-Te bulk has been extensively studied since 2005, achieving a PF of $\sim 30 \times 10^{-4} \text{ W m}^{-1} \text{ K}^{-2}$. However, to the authors' knowledge, only three studies have explored the thin-film format of Cu- or Ag-doped Bi-Sb-Te, all focusing on hot processing during deposition or post-annealing. These studies reported a maximum PF of $\sim 8 \times 10^{-4} \text{ W m}^{-1} \text{ K}^{-2}$, which is still lower than the highest PF ($10.5 \pm 0.5 \times 10^{-4} \text{ W m}^{-1} \text{ K}^{-2}$ for the 4Cu-Bi-Sb-Te alloy) achieved here with room-temperature processed materials. Surprisingly, despite the high metal content in our films (as compared to the literature in Figure 3 f), they retain some degree of semiconducting (i.e., thermoelectric) properties. This can be attributed to our specific fabrication technique - alternating sputtering and evaporation of sub-nanometer-thick, non-continuous layers - which leads to a heterogeneous structure comprising a mixture of semiconductor phases, metal-rich phases, and pure metal inclusions. It should be noted that in comparison to all relevant works listed in **Tables S1 & S3** (Supplementary Information), our work not only achieves excellent thermoelectric performance but also highlights a significant advancement in R2R large-scale manufacturing for large-area flexible materials. Most importantly, our technique demonstrates a unique patterning capability using SMT. Next, the parameters for the 4Cu-Bi-Sb-Te alloy will be utilized for the subsequent SMT study.

Page 6 Supplementary Information

Figure S3 (a) Summary of evaporated/sputtered Bi-Sb-Te materials from 2000 onwards (refer to Table S1 for detailed references).

2. The linear relationship between S and $\mu^{0.2}$ is not visible in Figure 3, the authors should mark it in the figure or give a more detailed explanation.

We have added a linear fit to Figure 3 h & i.

Page 12

Figure 3. (h) Seebeck coefficient vs (carrier mobility)^{0.2} (the lines indicate the linear fits for both the n-type and p-type regions); (i) Electrical resistivity vs (carrier concentration*mobility)⁻¹ (the lines indicate the linear fits for both the Ag and Cu groups).

Page 16

There were robust linear relationships with an R^2 value of ~ 0.9 between S and $\mu^{0.2}$ for both the n-type and p-type regions (Figure 3 h).

linear relationships, with a R^2 value of ~ 0.85 , were observed for both the Ag group and the Cu group.

3. While the effects of residual oil and how they are removed are discussed, the specific effects of different removal methods, such as electron beam or plasma treatment on thermoelectric properties can be further explored.

Additional analysis of the e-beam and plasma cleaning processes has been included. The plasma cleaning was performed using the HiTUS plasma system from Plasma Quest Ltd. The results were highly impressive, and we are preparing a follow-up paper to further discuss these findings.

Page 21

The Seebeck coefficient of the device (S_{TEG} , for a device with four pairs of thermoelectric strips) can be determined from the slope of the open-circuit voltage plot (see **Figure S3 b**, Supplementary Information). The e-beam-cleaned sample ($S_{TEG 3}$) showed a coefficient of 0.15 mV/K, while the plasma-cleaned sample ($S_{TEG 4}$) exhibited a slightly higher coefficient of 0.16 mV/K. These values, along with R_{TEG} , explain the higher power output observed in the plasma-cleaned device compared to the e-beam-cleaned device, as shown in Figure 4 (c). The extended 1-h e-beam exposure for the oil cleaning may have been too aggressive, potentially compromising the material's performance. However, as noted in our previous study [4], shorter e-beam cleaning times were insufficient to fully remove the residual oil. In contrast, plasma cleaning is rapid, taking only a few seconds, and is therefore more suitable for high-throughput R2R processes. This effectiveness is likely due to the interaction between the F-based oil and the plasma, where the active element F could play a

significant role. The plasma used in this study was generated using the HiTUS technique (Plasma Quest Ltd., UK), which allows for remote plasma generation and precise direction using electromagnets, making it highly compatible with R2R processes.

4. The detailed information about RTEG can only be founded in Experimental details part. The authors should provide an explanation about RTEG in the maintext. Otherwise, the discussion about resistance change might be confusing to the reader.

We have revised the text accordingly.

Page 20

The device resistance (R_{TEG}) of TEG with four pairs of thermoelectric strip/metal contact was measured using a two-probe method.

Page 22

R_{ETG} is the internal resistance of the thermoelectric generator device.

5. The application of this technology to wearable thermoelectric generators is impressive. However, the authors do not provide much performance information about the wearable thermoelectric device. Detailed performance data can better demonstrate practical application value of the device.

Thank you very much for these comments. Designing thin-film TEGs to be wearable is indeed a challenging question. While several studies have explored wrapping planar thin-film TEGs around rollers, our study is, to our knowledge, the first to propose a TEG wrist strap by integrating the wrapped TEG cell into a wrist strap design. We believe that with further research on this type of device, TEG cells will become smaller, wrist straps will become thinner, and more TEG cells can be incorporated, making the device more practical and capable of providing sufficient power for low-power wearable electronics. In this study, we have moved many results to the Supplementary Information (Figure S7 & S8 and COMSOL analysis, page 10-12) to focus the main text on the SMT technique and Ag- or Cu-containing Bi-Sb-Te materials.

We have revised the text accordingly.

Page 23 - 25

It should be noted that this voltage supply is generated by the temperature difference between the human skin and the ambient environment, making it a green, continuous, and reliable source. The voltage output of our device, which lies within the mV range as reported in [62], can be integrated with a converter and is sufficient to power a wristwatch. Additionally, this mV voltage level is suitable for use in wearables and implantables requiring low-level voltage supply sources [63-68].

In addition to material improvements, we believe the device output can be further enhanced through optimizing the integration process. **Figure S6** (e) in the Supplementary Information shows the resistance changes recorded after each fabrication step. The winding process does not significantly affect device resistance, likely due to the protective PDMS layer on top, despite involving oven processing. It should be noted that the rolling direction is perpendicular to the direction of the thermoelectric strip, which may slightly reduce the rolling-induced strain on the materials. However, integrating the device into the wrist frame increases resistance, highlighting the need for optimization at this step. Another significant resistance increase occurs during the connection of TEG cells in the frame, where Cu tape contacts are connected in series with Ag glue, and then cured in an oven (Step 6, **Figure S5**, Supplementary Information). The frame is subsequently embedded in PDMS and cured again (Step 7). These processes lead to a substantial rise in resistance to the M Ω range, which limits device output. Ideally, resistance should be within the k Ω range. For example, the resistance of a single thermoelectric pair in a flat condition is approximately 1 k Ω . Given that the prototype comprises 9 TEG cells, each with 25 thermoelectric pairs (totaling 225 pairs in the wrist strap), the ideal resistance would be around 225 k Ω , excluding contact resistance between TEG cells. Therefore, there is considerable potential to optimize the integration process, particularly the contacts between TEG cells. Additionally, the contact between the device and the heat source (i.e., human skin) should be improved. After PDMS molding (Step 7, **Figure S5**, Supplementary Information), direct contact between the TEG cells and the skin is prevented, limiting thermal contact and power output. Replacing the skin-side PDMS with a material of higher thermal conductivity could mitigate this issue. Furthermore, miniaturizing the TEG cells and optimizing the 3D-printed frame to integrate more cells into the strap could further enhance power output.

6. Considering the application environment of thermoelectric materials, the authors should perform long-term stability tests and provide test results.

We understand the reviewer's concern. Inorganic thermoelectric materials generally offer better long-term stability compared to organic materials. For these wearable applications, where the materials are exposed to relatively low temperatures (i.e., skin temperature) and do not undergo thermal cycling the stability, challenge may be less severe than more aggressive environments. All

our samples were stored under ambient conditions. We have conducted resistance measurements over two weeks to assess stability (Figure S9 Supplementary Information). Additionally, our previous work has demonstrated that Bi-Te-based materials fabricated using the same technique show negligible changes in thermoelectric performance over a year (DOI: 10.1016/j.surfcoat.2022.128826).

Page 12 Supplementary Information

Figure S9. Resistance variation of Bi-Sb-Te-based film over two weeks.

Reviewer #3 (Remarks to the Author):

This manuscript uses a research called Selective Metallization Technique (SMT), which is used to improve the performance of Bi-Sb in an industrial roll-to-roll (R2R) manufacturing environment. The authors successfully solved the adverse effects of fluorine used in the SMT oil process on Bi-Teki semiconductor materials by combining SMT with Physical Vapor Deposition (PVD) and co-evaporation of metal sources. The role of Cu and Ag as doping elements in Bi-Sb-Te films was studied, achieving the transition from n-type to p-type, and significantly improving the thermoelectric properties by adjusting the doping concentration. At the same time, the SMT-made Planar thermoelectric generators (TEGs) were fabricated into wearable devices, demonstrating the potential of this technology for practical applications. The method used is quite novel, and the experiments and results of the paper are fully displayed, but I think the author still needs to solve the following problems.

We appreciate the positive and constructive feedback.

1. The author selected the traditional Bi₂Te₃-based material as its excellent thermoelectric performance at room temperature. As we all known, both P-type and n-type samples can also be prepared by adjusting the ratio of Bi and Te. In this work, has the author ever thought of directly adjusting the ratio of Bi and Te to prepare P-type and n-type samples ? Is it necessary to introduce foreign doping elements (Ag and Cu)?

We understand the concern about introducing foreign doping elements. We have considered this question from three aspects:

(1) Bi-Sb-Te thin films: Extensive research has been conducted on Bi-Sb-Te thin films (see Table S1), focusing primarily on hot processing, which resulted in varying elemental ratios and improvements

in thermoelectric performance. However, metal-doped Bi-Sb-Te thin films are less studied compared to metal-doped Bi-Sb-Te bulk materials (see Table S3 Supplementary Information). To our knowledge, only four studies have investigated metal-doped Bi-Sb-Te thin films, and although hot processing was used, their performance remains lower than our film. In contrast, our room-temperature fabrication approach achieves performance comparable to that of bulk materials in Table S3.

(2) Sputtering-evaporation integration: We present a sputtering-evaporation integration technique for patterning in roll-to-roll processing. In our industrial roll-to-roll system, the evaporation source is fed in wire format and controlled by a feeder. However, materials such as Bi, Sb, and Te are typically supplied in pieces or pellets (e.g., from Kurt J. Lesker or Ted Pella), which lack the precision control of feeding as wire formats used in our evaporation process.

(3) Choice of Cu and Ag: The selection of Cu and Ag as doping elements is based on their cost-effectiveness and abundance, and many studies have investigated these two dopants in Bi-Sb-Te bulk materials.

We hope these explanations clarify our approach and the rationale behind our choices.

2. The author believes that the obtained p-type film has high performance, hence more literature data comparison is needed and presented in the manuscript. The high performance of Bi₂Te₃ thin film is an important reason for the selection of this material, but the highest power factor obtained by the author cannot fully reflect the advantages of this material system. Has the author conducted detailed research on performance optimization?

We have added more comparative data and extended the discussion (see Figure 3f, Figure S3a, and Tables S1, S2, and S3, Supplementary Information). Full thermoelectric characterization typically requires both power factor and thermal conductivity to calculate the figure of merit, which is common for bulk thermoelectrics. However, the power factor is often used as a primary measure in thin film thermoelectrics (see Tables S1 and S3, Supplementary Information). This is due to the challenges associated with measuring the thermal conductivity of thin films. While the Linseis Thin Film Analyzer can measure thermal conductivity of thin films, we found that depositing films on the analyzer's chip membrane can alter material properties, such as morphology, thickness, and composition, compared to films deposited on flexible polymer sheets. As a result, the measured thermal conductivity on the chip membrane differs from that on our flexible polymer substrate. We

are continuously seeking a suitable characterization technique for thin film thermoelectrics. In this study, we focus on highlighting the novelty of our roll-to-roll patterning technique.

Page 18 – 19

Compared to the Bi-Sb-Te film deposited under hot conditions (see **Figure S3 a**, Supplementary Information), the pristine Bi-Sb-Te film in this study exhibits a moderate *PF*. This reduction in performance could be attributed to the poor crystallinity of our film deposited at room temperature, which is constrained by the R2R processing. The most comparable study (i.e. room temperature, non-annealed, similar film thickness, see **Table S2**, Supplementary Information) had a *PF* of $0.9 \times 10^{-4} \text{ W m}^{-1} \text{ K}^{-2}$, which was approximately 10 times less than the highest *PF* obtained here ($10.5 \pm 0.5 \times 10^{-4} \text{ W m}^{-1} \text{ K}^{-2}$). This value was comparable to some hot-deposition/annealed Bi-Sb-Te. **Figure 3 (f)** compares Cu- or Ag-containing Bi-Sb-Te materials. The doped Bi-Sb-Te bulk has been extensively studied since 2005, achieving a *PF* of $\sim 30 \times 10^{-4} \text{ W m}^{-1} \text{ K}^{-2}$. However, to the authors' knowledge, only three studies have explored the thin-film format of Cu- or Ag-doped Bi-Sb-Te, all focusing on hot processing during deposition or post-annealing. These studies reported a maximum *PF* of $\sim 8 \times 10^{-4} \text{ W m}^{-1} \text{ K}^{-2}$, which is still lower than the highest *PF* ($10.5 \pm 0.5 \times 10^{-4} \text{ W m}^{-1} \text{ K}^{-2}$ for the 4Cu-Bi-Sb-Te alloy) achieved here with room-temperature processed materials. Surprisingly, despite the high metal content in our films (as compared to the literature in **Figure 3 f**), they retain some degree of semiconducting (i.e., thermoelectric) properties. This can be attributed to our specific fabrication technique - alternating sputtering and evaporation of sub-nanometer-thick, non-continuous layers - which leads to a heterogeneous structure comprising a mixture of semiconductor phases, metal-rich phases, and pure metal inclusions. It should be noted that in comparison to all relevant works listed in **Tables S1 & S3** (Supplementary Information), our work not only achieves excellent thermoelectric performance but also highlights a significant advancement in R2R large-scale manufacturing for large-area flexible materials. Most importantly, our technique demonstrates a unique patterning capability using SMT. Next, the parameters for the 4Cu-Bi-Sb-Te alloy will be utilized for the subsequent SMT study.

3. In Figure 5c, the author's intention cannot be clearly expressed in the Chinese film. It is recommended that the author give a more detailed description.

Thank you for the suggestion. We have added more details to the figure caption.

Page 25

The inset image shows the measurement setup. The red cables connect the device to a multimeter for voltage recording, while the blue cable connects to a thermocouple for measuring skin

temperature. The slight increase in the recorded voltage during the first 20 minutes in contact with the skin suggests the device is warming up, leading to a higher temperature difference and consequently a higher voltage output.

4. The author used Cu and Ag as doping elements in the preparation of Bi-Sb-Te films to achieve the transition from n-type to p-type, and significantly improved the thermoelectric performance by adjusting the doping concentration. The mechanism is explained incompletely. It is recommended that the author provide the physical mechanism of n-p transition and performance improvement.

We have added a schematic in Figure 2 and have included additional discussion in the text.

Page 10

Figure 2. (d) Electron shells of these elements with valence electrons; (e) Interstitial & Substitutional doping/alloying, including a visualization of the crystal structure and a band diagram showing anticipated acceptor/donor levels. Blue spheres represent the substitutional or interstitial atoms within the host lattice (grey spheres). Acceptor levels introduce more '+' (holes) into the valence band, while donor levels contribute more '-' (electrons) to the conduction band.

Page 10

Regarding the doping of Ag and Cu in the Bi-Sb-Te matrix, both interstitial and substitutional doping could occur, involving interstitial defects located between quintuple layers and substitution at Bi/Sb sites [19, 31, 33, 39], see Figure 2 (e).

Page 12 – 16

The following sections provide a detailed discussion regarding this type of transition phenomenon. ('n' in nAg and nCu denotes the feed-rate of the metal source during evaporation. A higher feed-rate indicates a greater metal content in the Bi-Sb-Te system).

Metal feed-rate of 0 - 1: At this stage, the material remains n-type. With few Ag and Cu dopants, the fraction of Bi and Sb remains almost constant (see Figure 2 c), but the Te fraction drops sharply, resulting in a decrease in carrier (electron) concentration. As a result, S increases in the n-type region due to the inverse relationship between n and S [40, 41]. As previously discussed, a varying-conductivity geometry (see Figure S1 f, Supplementary Information) could form heterostructure interfaces and induce an energy filtering effect [33] - a common method used to modify carrier properties and thereby enhance PF in thin-film thermoelectrics [31]. Carriers preferentially traverse the highly conductive regions, while the poorly doped regions with low conductivity act as energy filters at the boundaries. Carriers with low energy are blocked at the boundaries, whereas high-

energy carriers pass through, resulting in reduced n but increased μ , thereby increasing S (see Figure 3 a, d and e for the 1-metal case).

Metal feed-rate of 1 - 2: The material remains n-type. With an increased amount of metal source, interstitial doping could dominate, introducing donor levels close to the conduction band (see Figure 2 e), thus increasing n (Figure 3 d). Given the inverse relationship between carrier concentration and Seebeck performance [40, 41], the S decreases at this stage.

Metal feed-rate at 2 or 3: The semiconductor starts to transition from n-type to p-type, indicating that the majority of carriers are holes. This shift could be due to a change in the dominant doping mechanism from interstitial to substitutional, or the formation of other material phases. The coexistence of interstitial and substitutional doping has been observed in other material systems [42-44] and the doping type can be influenced by heat treatment [43]. Our fabrication approach includes heat treatment from the evaporation source. Although changes in doping type in the doped Bi-Sb-Te system have not been previously reported in the literature, it is known that Cu or Ag can be either interstitial or substitutional dopants in the Bi-Sb-Te system [39], see Figure 2 (e). It can be assumed that at low dopant concentrations, dopant atoms can easily occupy interstitial sites without significant lattice distortion. However, at higher dopant concentrations, interstitial sites become saturated, thus the crystal lattice cannot accommodate more atoms in the interstitial positions without significant lattice distortion. Consequently, the dopant atoms may start replacing host atoms, such as Bi or Sb, acting as substitutional dopants. In addition, the decreasing amounts of Sb and Te, as shown in Figure 2 (c) also affect the doping mechanism, if considering these elements as the original dopant sources in the pristine Te-excess Bi-Sb-Te system. It should be noted that other phases, such as the metal-dominated thermoelectric material Cu_2Te , could form [45-47] when a significant amount of metal source and heat treatment by the evaporation boat are involved. They might form an integrated system, such as $(\text{Cu-Te})(\text{Bi-Sb-Te})$ [45], or the Cu-Te phase could become a significant separate phase.

Metal feed-rate > 3: The material remains p-type as holes are the majority carriers. As the metal source increases, substitutional doping will introduce acceptor levels near the valence band (Figure 2 e). These levels accept electrons from the valence band, leaving more holes in the valence band that contribute to electrical conduction (Figure 3 b & d). As the hole concentration increases, the S decreases. At much higher metal content, a pure metal phase could form as inclusions within the thermoelectric matrix [48], which could significantly contribute to thermoelectric performance by enhancing S through energy filtering and/or reducing ρ via carrier channeling or injection. Alternatively, metal-rich thermoelectric phases e.g. Cu_2Te [49, 50], Cu_4Sb [51], and Ag_2Te [52-54]

could contribute to the performance of the alloy system. These phases may form due to the annealing effect induced by the substantial amount of hot metal source and the hot evaporation boat. The material would consist of multiple phases, such as crystalline/amorphous regions of Cu/Bi-Sb-Te/Cu₂Te/Cu₄Sb. This would introduce a significant number of boundaries, altering scattering mechanisms and affecting carrier transport properties. The energy filtering effect may occur, where the highly conductive phases decrease ρ while less conductive regions serve as energy filters, increasing S . Hence, the formation of metal-rich phases and pure metal inclusions can have a complex yet potentially beneficial effect on thermoelectric performance, resulting in a notably high PF for the 4Cu sample (see Figure 3 c).

Based on the analysis above, the transition from interstitial to substitutional doping could explain the n-type-to-p-type transition observed at a metal feed-rate at 2 or 3. This transition likely leads to a shift in the Fermi level, which initially rises and then falls as the dopant concentration increases [55]. This analysis is based on a simplified model of the doping mechanism within the Bi-Sb-Te matrix (Figure 2 e). However, the actual situation is more complex, particularly due to our room-temperature fabrication process (resulting in poor crystallinity, as confirmed in our previous works [56, 57]), annealing effects from the hot evaporation boat/vapor source, and the heterogeneous structure during layer-by-layer deposition. Additionally, metal-rich thermoelectric phases could form due to the significant amount of metal source involved. These factors result in variations in elemental composition, grain size and crystallinity as well as the coexistence of multiple phases, creating a varying-conductivity geometry.

Reviewer #4 (Remarks to the Author):

The authors proposed an innovative method on fabricating flexible Bi-Sb-Te thermoelectric (TE) films, which is based on in-line roll-to-roll patterning, the sputtering of BiSbTe target and the evaporation of metal. In addition, this work demonstrated that novel single-leg TE units could be fabricated by the roll-up process, which are effective in harvesting body heat for applications in wearable electronics. In general, this work represented an importance advancement on flexible TE devices, especially the novel fabrication process. However, before the final judgement from the editors, there are several important issues that should be addressed.

We appreciate the reviewer's comments.

(1) Large scale fabrication?

The following content are listed in the manuscript, including “1-meter wide polymer web”, “high throughput of 25m min⁻¹ in-line speed”, “The sputtering condition on a three-inch target”, and others. These information describe the parameters of fabrication apparatus, but not for clarifying the large scale fabrication.

What is the size of fabricated films in one time, and what is the thickness of films? In order to clearly elucidate the large scale fabrication, optical images with scale bar, the uniformity of film thickness, and the phase structure in the large area should be provided.

Thank you very much for this suggestion. We have added a Supplementary Video demonstrating large-scale manufacturing using our roll-to-roll facility. Additionally, Figure 1 has been modified, and we have included more details on the uniformity of the fabricated materials in the text.

Supplementary Video

Page 4

This technique enables high-throughput in-line patterning in an industrial R2R setting, achieving micron resolution and high-density arrays, as demonstrated in the Supplementary Information Video.

Page 5

Figure 1. *In-line patterning of flexible thin films at high throughput by simultaneous sputtering of functional materials and evaporation of metal in an SMT. In evaporation, a feed-rate of 20 corresponded to an in-line speed of 8 cm min⁻¹ for ~1 mm diameter metal wires. The coating drum was rotated at an in-line speed of 25 m min⁻¹. The images show the setup of flexography and the coating drum, demonstrating the large-scale capability for achieving micro-resolution and high-density arrays.*

Page 30

The stability of the material's resistivity was characterized using a custom in-house four-point probe system at 15 different locations across three identical samples. These samples were taken from three different locations on the coating drum (radius: 0.3 m, width: 0.8 m), randomly selected from a total sample size of 1.8 m × 15 cm. Although the coating drum has a width of 0.8 m, we selected samples from the middle region (15 cm wide) due to the comparatively small diameter of the sputtering target. As shown in **Figure S9** (Supplementary Information), the material exhibited negligible resistance change over two weeks of storage under ambient conditions. Additionally, the error bars indicate uniformity across the coating drum. Batch-to-batch variation was assessed

through three identical material fabrication batches, revealing a 3% variation in Seebeck characteristics. For the device, the thermoelectric pair exhibited a resistance variation of approximately 6%. These results confirm the uniformity of the films fabricated using our large-scale manufacturing technique.

Page 2 Supplementary Information

Figure S1 (b) The film thickness (the error bar represents an average from ten different locations from the 1.8 m × 15 cm sample on the coating drum)

(2) The origin of TE properties of Bi-Sb-Te films?

The authors claimed in the introduction section that the doping of Cu and Ag can respectively lead to n-type and p-type TE properties of Bi₂Te₃-based materials, which was generally accepted by the TE community. However, the results in this work is totally different, and the discussions based on defect engineering is vague without solid evidences from phase structure and microstructure. In addition, the fabricated films obtained very low electrical conductivity, which shows a big discrepancy from the original Bi_{0.5}Sb_{1.5}Te₃ target. The authors should establish the correlations of composition—structure—properties for the fabricated films.

We have expanded the discussion on the doping mechanism. It has been reported that Cu and Ag can act as both interstitial and substitutional defects in the Bi-Sb-Te system, and then function as either acceptor or donor defects (DOI: 10.1016/j.jallcom.2017.08.166). Our results confirm that both scenarios can occur, with the dominant carrier type and concentration varying accordingly. Additional detailed discussion has been incorporated into the text.

We acknowledge that the performance of our pristine Bi-Sb-Te film is not optimal. However, it is comparable to the performance of room-temperature sputtered Bi-Sb-Te (DOI: 10.1016/j.matpr.2020.04.752), see Table S2 Supplementary Information. We attribute the lower performance of our pristine Bi-Sb-Te film to its poorer crystallinity, which is a consequence of our focus on room-temperature sputtering. While hot processing deposition yields much higher performance, it is not compatible with roll-to-roll manufacturing process in our case.

Page 5 in Supplementary Information

Table S2. Comparison of room-temperature deposited Bi-Sb-Te film (referenced from Table S1).

The following sections provide a detailed discussion regarding this type of transition phenomenon. (' n ' in n_{Ag} and n_{Cu} denotes the feed-rate of the metal source during evaporation. A higher feed-rate indicates a greater metal content in the Bi-Sb-Te system).

Metal feed-rate of 0 - 1: At this stage, the material remains n-type. With few Ag and Cu dopants, the fraction of Bi and Sb remains almost constant (see Figure 2 c), but the Te fraction drops sharply, resulting in a decrease in carrier (electron) concentration. As a result, S increases in the n-type region due to the inverse relationship between n and S [40, 41]. As previously discussed, a varying-conductivity geometry (see **Figure S1 f**, Supplementary Information) could form heterostructure interfaces and induce an energy filtering effect [33] - a common method used to modify carrier properties and thereby enhance PF in thin-film thermoelectrics [31]. Carriers preferentially traverse the highly conductive regions, while the poorly doped regions with low conductivity act as energy filters at the boundaries. Carriers with low energy are blocked at the boundaries, whereas high-energy carriers pass through, resulting in reduced n but increased μ , thereby increasing S (see Figure 3 a, d and e for the 1-metal case).

Metal feed-rate of 1 - 2: The material remains n-type. With an increased amount of metal source, interstitial doping could dominate, introducing donor levels close to the conduction band (see Figure 2 e), thus increasing n (Figure 3 d). Given the inverse relationship between carrier concentration and Seebeck performance [40, 41], the S decreases at this stage.

Metal feed-rate at 2 or 3: The semiconductor starts to transition from n-type to p-type, indicating that the majority of carriers are holes. This shift could be due to a change in the dominant doping mechanism from interstitial to substitutional, or the formation of other material phases. The coexistence of interstitial and substitutional doping has been observed in other material systems [42-44] and the doping type can be influenced by heat treatment [43]. Our fabrication approach includes heat treatment from the evaporation source. Although changes in doping type in the doped Bi-Sb-Te system have not been previously reported in the literature, it is known that Cu or Ag can be either interstitial or substitutional dopants in the Bi-Sb-Te system [39], see Figure 2 (e). It can be assumed that at low dopant concentrations, dopant atoms can easily occupy interstitial sites without significant lattice distortion. However, at higher dopant concentrations, interstitial sites become saturated, thus the crystal lattice cannot accommodate more atoms in the interstitial positions without significant lattice distortion. Consequently, the dopant atoms may start replacing host atoms, such as Bi or Sb, acting as substitutional dopants. In addition, the decreasing amounts of Sb and Te, as shown in Figure 2 (c) also affect the doping mechanism, if considering these elements as the original dopant sources in the pristine Te-excess Bi-Sb-Te system. It should be noted that other

phases, such as the metal-dominated thermoelectric material Cu_2Te , could form [45-47] when a significant amount of metal source and heat treatment by the evaporation boat are involved. They might form an integrated system, such as $(\text{Cu-Te})(\text{Bi-Sb-Te})$ [45], or the Cu-Te phase could become a significant separate phase.

Metal feed-rate > 3: The material remains p-type as holes are the majority carriers. As the metal source increases, substitutional doping will introduce acceptor levels near the valence band (Figure 2 e). These levels accept electrons from the valence band, leaving more holes in the valence band that contribute to electrical conduction (Figure 3 b & d). As the hole concentration increases, the S decreases. At much higher metal content, a pure metal phase could form as inclusions within the thermoelectric matrix [48], which could significantly contribute to thermoelectric performance by enhancing S through energy filtering and/or reducing ρ via carrier channeling or injection. Alternatively, metal-rich thermoelectric phases e.g. Cu_2Te [49, 50], Cu_4Sb [51], and Ag_2Te [52-54] could contribute to the performance of the alloy system. These phases may form due to the annealing effect induced by the substantial amount of hot metal source and the hot evaporation boat. The material would consist of multiple phases, such as crystalline/amorphous regions of $\text{Cu/Bi-Sb-Te/Cu}_2\text{Te/Cu}_4\text{Sb}$. This would introduce a significant number of boundaries, altering scattering mechanisms and affecting carrier transport properties. The energy filtering effect may occur, where the highly conductive phases decrease ρ while less conductive regions serve as energy filters, increasing S . Hence, the formation of metal-rich phases and pure metal inclusions can have a complex yet potentially beneficial effect on thermoelectric performance, resulting in a notably high PF for the 4Cu sample (see Figure 3 c).

Based on the analysis above, the transition from interstitial to substitutional doping could explain the n-type-to-p-type transition observed at a metal feed-rate at 2 or 3. This transition likely leads to a shift in the Fermi level, which initially rises and then falls as the dopant concentration increases [55]. This analysis is based on a simplified model of the doping mechanism within the Bi-Sb-Te matrix (Figure 2 e). However, the actual situation is more complex, particularly due to our room-temperature fabrication process (resulting in poor crystallinity, as confirmed in our previous works [56, 57]), annealing effects from the hot evaporation boat/vapor source, and the heterogeneous structure during layer-by-layer deposition. Additionally, metal-rich thermoelectric phases could form due to the significant amount of metal source involved. These factors result in variations in elemental composition, grain size and crystallinity as well as the coexistence of multiple phases, creating a varying-conductivity geometry.

(3) Uniformity of TE properties?

For large scale applications, the uniformity of TE properties should be demonstrated for fabricated films.

Thank you for the suggestion. We have added more results regarding the uniformity of the fabricated materials.

Page 30

The stability of the material's resistivity was characterized using a custom in-house four-point probe system at 15 different locations across three identical samples. These samples were taken from three different locations on the coating drum (radius: 0.3 m, width: 0.8 m), randomly selected from a total sample size of 1.8 m × 15 cm. Although the coating drum has a width of 0.8 m, we selected samples from the middle region (15 cm wide) due to the comparatively small diameter of the sputtering target. As shown in Figure S9 (Supplementary Information), the material exhibited negligible resistance change over two weeks of storage under ambient conditions. Additionally, the error bars indicate uniformity across the coating drum. Batch-to-batch variation was assessed through three identical material fabrication batches, revealing a 3% variation in Seebeck characteristics. For the device, the TE pair exhibited a resistance variation of approximately 6%. These results confirm the uniformity of the films fabricated using our large-scale manufacturing technique.

(4) The reason for too low device performances?

Does the too low device performances come from too thin thickness? Since micro-cracks could be introduce in inorganic films/devices during the fabrication process, the detection and the influence of cracks should be discussed in the manuscript. Also, a roll-up process was used to integrate the single-leg TE unit, and a very large internal resistance was reported for the integrated device, unfavorable for high performance TE devices. The influence of the roll-up process on the TE performance, the reduction of internal resistance, and the enhancement of device performance should be added in order for verifying the effectiveness of the innovative method proposed by the authors.

Thank you very much for this suggestion.

A thickness of approximately 100 nm is desired using our technique because: (1) Our previous work has shown that thermoelectric performance is optimal in this thickness range (DOI:

10.1016/j.surfcoat.2020.125393). (2) The SMT technique used in this study requires a specific thickness range; if the film is too thick, the oil pattern used in SMT may be insufficient, which can adversely affect the patterning process.

Thicknesses at this level generally do not exhibit cracking issues (DOI: 10.1016/S1359-6454(02)00254-9). Our SEM images, shown in Figure S1 of the supplementary information, confirm this. Additionally, a large MicroXAM image provided below further demonstrates that the film is crack-free.

In terms of device integration, the roll-up process is indeed suboptimal. We have included data on the changes in internal resistance of the device to discuss this issue.

Page 9 in Supplementary Information

Figure S6. Experimental steps illustrated in Figure S5: (a) Step 2 – Cu wire connection using Ag paste and cured in an oven; (b) Step 3 – Blade coating of PDMS on top, followed by oven curing; (c) Step 4 – Manual winding; (d) Step 6 – Integration into the frame; (e) Resistance change after each step, averaged from 20 samples. The resistance result at Step 6 is from a single TEG cell in the frame, as shown in (d). After completing Step 8 in Figure S5, the resistance significantly increased to the MΩ range.

Page 24

In addition to material improvements, we believe the device output can be further enhanced through optimizing the integration process. Figure S6 (e) in the Supplementary Information shows the resistance changes recorded after each fabrication step. The winding process does not significantly affect device resistance, likely due to the protective PDMS layer on top, despite involving oven processing. It should be noted that the rolling direction is perpendicular to the

direction of the thermoelectric strip, which may slightly reduce the rolling-induced strain on the materials. However, integrating the device into the wrist frame increases resistance, highlighting the need for optimization at this step. Another significant resistance increase occurs during the connection of TEG cells in the frame, where Cu tape contacts are connected in series with Ag glue, and then cured in an oven (Step 6, **Figure S5**, Supplementary Information). The frame is subsequently embedded in PDMS and cured again (Step 7). These processes lead to a substantial rise in resistance to the M Ω range, which limits device output. Ideally, resistance should be within the k Ω range. For example, the resistance of a single thermoelectric pair in a flat condition is approximately 1 k Ω . Given that the prototype comprises 9 TEG cells, each with 25 thermoelectric pairs (totaling 225 pairs in the wrist strap), the ideal resistance would be around 225 k Ω , excluding contact resistance between TEG cells. Therefore, there is considerable potential to optimize the integration process, particularly the contacts between TEG cells. Additionally, the contact between the device and the heat source (i.e., human skin) should be improved. After PDMS molding (Step 7, **Figure S5**, Supplementary Information), direct contact between the TEG cells and the skin is prevented, limiting thermal contact and power output. Replacing the skin-side PDMS with a material of higher thermal conductivity could mitigate this issue. Furthermore, miniaturizing the TEG cells and optimizing the 3D-printed frame to integrate more cells into the strap could further enhance power output.

Response to reviewers

We appreciate the reviewers' comments on our work. We have made some modifications to clarify the narrative, and we hope it now comes across more clearly. Below is a detailed response to each point. The reviewers' comments are in bold, our responses and the locations of changes in the manuscript are highlighted in green, and the newly revised text is highlighted in yellow.

Reviewer #4 (Remarks to the Author):

The authors response to my queries and also to other reviewer's questions, including the phase structure, the printing details, the data analysis and device performances. Although the printing process is impressive for printing metal and Bi-Sb-Te, the apparatus can not print thick films, not beneficial for thermoelectric applications. The reviewer could not suggest the publication of this manuscript to this prestigious journal with two major reasons below.

1. The analysis of thermoelectric performance strongly relies on the understanding of chemical composition, phase structure and microstructure of materials being studied, since these parameters will lead to remarkable change on every thermoelectric transport parameters and carrier transport parameters. The authors give the speculation that the films could be Cu/Ag doped or Cu/Ag intercalated BiSbTe, or Cu/Ag contained BiSbTe, or Cu-Te/Ag-Te mixed BiSbTe, but without direct experimental evidences. XRD show no obvious peak from BiSbTe. In addition, the thermoelectric properties deviate remarkably from the BiSbTe bulk and thin films. Thus, the reviewer could not accept the discussion and the conclusion on the thermoelectric transport, due to too many possibilities if the chemical composition and microstructure is not precisely determined.

We appreciate the reviewer's comments and understand the concern raised.

The thermoelectric properties of our film fall within the range of Ag- (or Cu-) Bi-Sb-Te bulk and thin films (see Figure 3 f), as well as within the range for Bi-Sb-Te films (see Figure S3 a, Supplementary Information). It is important to note that our film is fabricated at room temperature to ensure compatibility with roll-to-roll processing for industrial-scale manufacturing. This choice may involve some trade-offs compared to hot-processed Bi-Sb-Te films, as shown in Figure S3 a (Supplementary Information).

In terms of the materials microstructure, the Cu-Bi-Sb-Te structure is a composite with alternating layers of Cu and Bi-Sb-Te, each layer measuring subnanometer in thickness. This configuration creates a uniform Cu-Bi-Sb-Te composite, so distinct clusters of metal and Bi-Sb-Te regions are not expected to appear, as confirmed by EDX mapping (Figure S1 a, Supplementary Information). The

chemical composition of the film is shown in Figure 2 c, and its surface morphology can be seen in Figure S1 a (Supplementary Information) and Figure 2 a. We have also included EDX mapping in Figure S1 a to illustrate the uniformity of the elemental distribution.

Our XRD analysis for phase identification is indeed limited, and we have added further discussion to address this. Unlike micron-thick films, nanometer-thick films are challenging to analyze with XRD, even when grown on silicon wafers, as their thinness prevents sufficient signal collection. In our previous work, we confirmed the Bi-Te material phase in 1- μm thick films using the same fabrication technique (<https://doi.org/10.1016/j.tsf.2020.138311>). In our roll-to-roll setup, the ultra-thin films enable high deposition throughput, helping to keep manufacturing costs low. Our approach is focused on using SMT for industrial-scale manufacturing of nanometer-thick patterns, which traditional printing techniques struggle to achieve. Thermoelectric materials serve as a representative example for our SMT technique because nanostructured thermoelectric films hold promise for enhancing thermoelectric performance (<https://doi.org/10.1038/asiamat.2010.138>; <https://doi.org/10.1016/j.pnsc.2012.11.011>).

Page 16

We can confirm the presence and uniform distribution of metal, Bi, Sb, and Te elements through SEM/EDX (see Figure S1 a, Supplementary Information), however, identification of the multiple possible phases is not feasible in our case because the film thickness is insufficient for XRD detection (see Figure S2, Supplementary Information). This limitation is further compounded by the room-temperature R2R deposition technique, which restricts film crystallinity. We selected this thickness range because rapid R2R thin-film deposition becomes less favorable for thicker films; the very thin films have the manufacturing advantage of high deposition throughputs, to keep down the manufacturing cost. The nanometer thickness of some functional materials, for example thermoelectric materials [7, 8], is particularly advantageous for enhancing performance. Our prior work has demonstrated that films of this thickness can achieve good thermoelectric performance [57]. This thickness range is compatible with the SMT technique, which is constrained by the thickness of the oil patterns, although the oil thickness can be adjusted in flexography.

Page 7

In the form of the sub-nanometer-thick alternating layers, the metal and Bi-Sb-Te regions are indistinguishable due to their uniform distribution, as confirmed by energy-dispersive X-ray spectroscopy (EDX) mapping (see Figure S1 a, Supplementary Information).

Figure S1. (a) SEM and EDX images

2. The authors claim the optimal thickness of films is around 100 nm for the adopted process. This limits the thermoelectric application arising from too large internal resistance of fabricated films.

We apologize for any confusion regarding film thickness. Our SMT technique is capable of producing thicker films, with the thickness depending on the properties of the oil used, such as viscosity. By printing thicker oil patterns, a correspondingly thicker functional film can be achieved. However, the primary goal of our SMT technique is to create ultra-thin (nano-scale) patterns. For micron-thick patterns, traditional printing technique can achieve this. Unlike traditional direct printing techniques, our SMT offers greater material versatility by integrating with physical vapor deposition (PVD). This allows SMT to pattern a wide range of materials compatible with PVD, including metals, functional materials, and insulators.

Our focus is on nanometer-scale thicknesses because (1) high-speed roll-to-roll processing is not suited for thicker films, (2) nanometer-scale materials often exhibit superior performance and consumes less semiconductor material, and (3) this thickness is optimal for SMT patterning. We have added further details on this in the manuscript.

Page 16

We selected this thickness range because rapid R2R thin-film deposition becomes less favorable for thicker films; the very thin films have the manufacturing advantage of high deposition throughputs, to keep down the manufacturing cost. The nanometer thickness of some functional materials, for example thermoelectric materials [7, 8], is particularly advantageous for enhancing performance. Our prior work has demonstrated that films of this thickness can achieve good thermoelectric performance [57]. This thickness range is compatible with the SMT technique, which is constrained by the thickness of the oil patterns, although the oil thickness can be adjusted in flexography.

Page 4

This technique enables high-throughput in-line patterning of nanometer-thick metal and functional materials in an industrial R2R setting.

Page 2

Recent advances in the Selective Metallization Technique (SMT) [2-4] show promise over conventional patterning techniques, such as printing, which is generally restricted to micrometer-range thicknesses [5, 6]. However, certain materials, like thermoelectric materials, benefit from nanostructure to optimise performance [7, 8], and nanometer-thick thin films reduce concerns related to material cracking [9], which is crucial for flexible electronics applications.

Response to reviewers

We thank the reviewers for their valuable feedback and suggestions. We have addressed all concerns in the revised manuscript. The reviewers' comments are in bold, our responses and changes are highlighted in green, and the revised text is in yellow.

Reviewer #4 (Remarks to the Author):

As I mentioned in previous comments, this manuscript is impressive on the fabrication process as well as on the large scale printing of thermoelectric films. The drawback of this technique is the low crystallinity of films as well as the thickness control.

Since other reviewers provided positive evaluations on this manuscript, I would respect their opinions. To adequately reveal the importance of this manuscript (also not to confuse the broad readership), the reviewer will suggest the following two points that should be addressed before the final judgement from the editor.

1. The composition of films should be written as Bi-Sb-Te based composite films, which is more suitable to describe the structure feature.

We have revised the manuscript accordingly to incorporate your suggestion.

Page 1

The room-temperature-deposited material system exhibits significantly enhanced thermoelectric performance and facilitates an n-type-to-p-type transition in the Cu- or Ag-containing Bi-Sb-Te-based composite film.

Page 7

The elemental composition of the Bi-Sb-Te-based composite films was determined through EDX.

Page 9

Figure 2. (c) Elemental composition from EDX-SEM of Bi-Sb-Te composite films

Page 26

the Cu- or Ag-containing Bi-Sb-Te-based composite materials exhibited an interesting n-type-to-p-type transition.

2. The flexible and thin thermoelectric films are most suitable for high precision temperature or radiation detection. The reviewer suggests the authors to highlight this point, since high resistance films are useless on thermoelectric power generation applications.

We have incorporated this information into the manuscript as suggested.

Page 21

High-resistance flexible thin-film thermoelectrics are suitable for high-precision temperature or radiation detection. Here, we explore their potential application as power generators.